# OpenReview forum: "AdLift: Lifting Adversarial Perturbations to Safeguard 3D Gaussian Splatting Assets Against Instruction-Driven Editing"
_ICLR.cc/2026/Conference — Submitted to ICLR 2026_

### Official Review · Reviewer_nZzV · 2025-10-25

**Soundness:** 3
**Presentation:** 3
**Contribution:** 4
**Rating:** 6
**Confidence:** 4

**Summary:**

This paper presents the first approach to safeguarding 3D Gaussian Splatting (3DGS) assets against malicious edits, including both 2D image editing and local/global 3D manipulations, by lifting adversarial perturbations into the 3D space. The core challenges lie in maintaining protection effectiveness across multiple views and balancing imperceptibility with robustness. AdLift addresses these issues via a Lifted PGD strategy that avoids direct gradient flow from the target model to the 3D Gaussians; instead, it truncates gradients at the 2D rendered image and lifts them back to the 3D representation.

**Strengths:**

1. The first work to extend adversarial image protection from 2D image editing to 3DGS representations.
2. Comprehensive evaluation spanning both 2D image synthesis attacks and local/global 3DGS editing scenarios.
3. Clearly written and well-organized, making the technical contributions easy to follow.

**Weaknesses:**

1. While the authors argue that soft-constrained baselines (e.g., GuardSplat) or direct optimization with hard constraints on attributes are fundamentally limited, the paper lacks sufficient experimental evidence to support this claim. Although adversarial loss and CLIP score trends are reported under varying PSNR values, there is a lack of concrete editing results to validate whether these numerical differences meaningfully translate to protection effectiveness.
2. AdLift is designed to generate view-consistent adversarial perturbations across multiple viewpoints. However, the paper provides only qualitative evidence (e.g., visual comparisons) to support this claim, with no quantitative metrics or consistency measurements to validate the view-generalizability of perturbations.
3. The rationale behind duplicating the original Gaussians and optimizing the copies is not clearly justified. This design choice may raise concerns in terms of scalability and efficiency, especially in complex scenes.

Note: Weaknesses 1-3 correspond directly to Questions 1-3.

**Questions:**

1. While the paper claims that soft-constrained baselines (e.g., GuardSplat) and direct optimization with hard constraints on attributes are less effective, the experimental evidence supporting this claim remains limited. It would be helpful to see qualitative editing results comparing AdLift and GuardSplat at the same PSNR levels.
For example, in Figure 10, GuardSplat achieves a higher CLIP score at the same PSNR, but it is unclear whether this corresponds to weaker editing resistance in practice. Similarly, it is unclear whether the perturbed images in Figure 11 are matched in noise level. Including additional results where GuardSplat or hard-constraint baselines are evaluated under similar perceptual distortion (e.g., PSNR) with AdLift would strengthen the argument.
Lastly, a brief explanation for why the adversarial loss does not decrease under these soft-constrained setups—whether due to gradient misalignment, saturation, or other structural factors—would help clarify the underlying limitation.
2. Can the authors provide quantitative evidence of view consistency? For instance, how does AdLift compare to prior watermarking approaches like GaussianMarker or GuardSplat in terms of perturbation alignment across views?
3. Does duplicating the Gaussians lead to significant memory overhead, particularly for dense scenes where the number of primitives is already high? A discussion or analysis of the scalability implications would be helpful.
4. Can the authors report additional evaluation metrics such as FID or Precision, in addition to CLIP score? Prior 2D image protection works like PhotoGuard[1] and AdvDM[2] typically report FID and Precision, which are widely adopted and allow for better comparability. It would also be helpful to provide these metrics for the editing results of the baseline methods discussed in Q1.
5. In Figure 8, the qualitative results of local 3DGS editing (e.g., the bear scene) show only marginal visual differences between the unprotected and protected assets. In contrast, Table 2 reports that the proposed AdLift-ST achieves substantially stronger protection performance (e.g., lower CLIPd) compared with Instruct-GS2GS and DGE (Global). How does AdLift-ST yield such strong quantitative gains despite the visual outputs in Figure 8 appearing relatively similar on the bear scene? Additional explanation would help clarify the effectiveness of local protection.


[1] Salman, Hadi, et al. "Raising the cost of malicious ai-powered image editing." arXiv preprint arXiv:2302.06588 (2023).

[2] Liang, Chumeng, et al. "Adversarial example does good: Preventing painting imitation from diffusion models via adversarial examples." arXiv preprint arXiv:2302.04578 (2023).

---

> ### Author Response · Authors · 2025-11-26
> **Response to Reviewer nZzV (Part 1/5)**
>
> Dear Reviewer `nZzV`,
>
> Thanks for your positive opinions. We address your concerns as follows. If anything remains unclear, please do not hesitate to contact us.
>
> > **W1&Q1: While the paper claims that soft-constrained baselines (e.g., GuardSplat) and direct optimization with hard constraints on attributes are less effective, the experimental evidence supporting this claim remains limited. It would be helpful to see qualitative editing results comparing AdLift and GuardSplat at the same PSNR levels. A brief explanation for why the adversarial loss does not decrease under these soft-constrained setups—whether due to gradient misalignment, saturation, or other structural factors—would help clarify the underlying limitation.**
>
> **Response 1:** Thank you for the valuable suggestions. To further support our claim, we provide additional results including: (i) comparisons with GuardSplat under matched PSNR or protection levels, and (ii) comparisons with additional baselines under matched PSNR levels.
>
> **Comparison with GuardSplat at matched PSNR or protection levels:**
> - **Experimental setup.** We provide both quantitative and qualitative comparisons between AdLift and GuardSplat [D1] under matched PSNR or protection levels. The quantitative results are reported in **Table D1-D2**, which include:
>   - (i) comparisons under similar PSNR levels (denoted as GuardSplat-VU1 vs. AdLift-VU) and
>   - (ii) comparisons under similar anti-editing performance with matched perceptual quality (denoted as  GuardSplat-VU2 vs. AdLift-VU).
>
>   We also report the adversarial loss values along with the additional three editing metrics (FID, F$_{1/8}$, and F$_8$) as you suggested. More detailed demonstration of the three metrics are shown in **Response 4**. Besides, qualitative editing results comparing AdLift and GuardSplat at the same PSNR levels are shown in **Figure 28-29** in the revised manuscript.
>
> - **Results.** The quantitative and qualitative results consistently confirm that AdLift better balances imperceptibility and protection effectiveness: under matched PSNR levels AdLift delivers stronger editing resistance, and under matched protection strength it maintains higher perceptual quality compared to GuardSplat-based baselines.
> - **Adversarial loss for soft-constraint methods.** To clarify, under soft-constrained optimization the adversarial loss can indeed increase for the untargeted attack (where a larger loss is desirable). Actually, the core issue of soft-constrains lies in balancing attack effectiveness and visual quality. Even with careful tuning of the adversarial loss weight, the soft constraint provides weaker control over perceptual fidelity compared to our hard-constrained method. As a result, perturbations become more visible. For this reason, we adopt a hard-constrained design in our AdLift, which ensures a better balance between effective protection and visual quality.
>
>
> **Comparison with more baselines at the matched PSNR level**:
> - **Experimental setup.** We additionally include several potential baselines by adapting representative 3DGS watermarking methods into an adversarial training regime, and compare them against AdLift under matched perceptual quality. Specifically, we include the following variants:
>   - GuardSplat-(SH): Perturb only the *Spherical-Harmonic* (SH) features of all Gaussians, following the original GuardSplat [D1] design.
>   - GuardSplat-(PC): Perturb *Positions* and *Covariance* features of all Gaussians using GuardSplat [D1].
>   - GuardSplat-(Full): Perturb all features of all Gaussians using GuardSplat [D1], including *Position*, *Covariance*, *Opacity*, and *SH features*.
>   - GaussianMarker [D2]: Split low-uncertainty Gaussians and optimize the newly added Gaussians to encode perturbations.
>
>   For each method, we evaluate two adversarial objectives, untargeted VAE loss (VU) and targeted VAE loss (VT). We preserve each method's original fidelity-preserving loss but replace original watermark decoding loss with our adversarial objective so that all methods are optimized toward the same protection goal.
> - **Results.** The quantitative results are provided in **Table D3** (training views) and **Table D4** (novel views). We report protection performance under similar perceptual quality (similar PSNR). We also include representative qualitative comparisons in **Figure 30** (training views) and **Figure 31** (novel views) in the revised manuscript. From the results, AdLift consistently achieves stronger anti-editing performance under similar invisibility constraints.

---

> ### Author Response · Authors · 2025-11-26
> **Response to Reviewer nZzV (Part 2/5)**
>
> **Table D1:** Comparison with GuardSplat at the same PSNR-level or anti-editing-level (training views).
> |Scene| Method|SSIM ($\uparrow$)|PSNR ($\uparrow$)|LPIPS ($\downarrow$)|CLIP_d ($\downarrow$)|CLIP_s ($\downarrow$)|AdvLoss ($\uparrow$)|FID ($\uparrow$)|F_1/8 ($\downarrow$)|F_8 ($\downarrow$)
> |:-:|:-:|:-:|:-:|:-:|:-:|:-:|:-:|:-:|:-:|:-:|
> |Face| No Protection|0.9381|32.6436|0.0668|0.1685|0.8852|--|22.0270|0.9958|0.9947|
> || GuardSplat-VU1|0.9348|31.7765|0.0802|0.1649|0.8769|5.4381|26.2044|0.9945|0.9952|
> || GuardSplat-VU2|0.4855|7.0444|0.6414|0.1479|0.7387|51.6294|126.1376|0.7694|0.6478|
> || AdLift*-VU|0.7001|28.3963|0.3774|0.1554|0.7620|29.2730|88.5865|0.8954|0.8742|
> |Fangzhou| No Protection|0.9218|32.1021|0.1093|0.1937|0.9080|--|23.7996|0.9817|0.9795|
> || GuardSplat-VU1|0.8596|23.4021|0.2126|0.1874|0.8380|18.4520|53.6308|0.9248|0.9653|
> || GuardSplat-VU2|0.3850|6.4662|0.6600|0.1727|0.7432|58.1082|121.3719|0.6165|0.4988|
> || AdLift*-VU|0.6706|28.3497|0.3767|0.1781|0.7900|29.4458|84.2180|0.8030|0.7749|
>
> **Table D2:** rison with GuardSplat at the same PSNR-level or anti-editing-level (novel views).
> |Scene| Method|SSIM ($\uparrow$)|PSNR ($\uparrow$)|LPIPS ($\downarrow$)|CLIP_d ($\downarrow$)|CLIP_s ($\downarrow$)|AdvLoss ($\uparrow$)|FID ($\uparrow$)|F_1/8 ($\downarrow$)|F_8 ($\downarrow$)
> |:-:|:-:|:-:|:-:|:-:|:-:|:-:|:-:|:-:|:-:|:-:|
> |Face| No Protection|0.8590|26.0390|0.1292|0.1704|0.8730|--|76.2210|0.9882|0.9835|
> || GuardSplat-VU1|0.8552|25.7744|0.1442|0.1654|0.8666|6.3217|83.8178|0.9846|0.9875|
> || GuardSplat-VU2|0.4328|7.3565|0.6576|0.1501|0.7361|47.7983|176.4602|0.7253|0.5904|
> || AdLift*-VU|0.6367|25.6115|0.4018|0.1586|0.7786|20.2232|129.8172|0.8807|0.9279|
> |Fangzhou| No Protection|0.9010|30.2873|0.1161|0.1938|0.9053|--|51.7357|0.9905|0.9913|
> || GuardSplat-VU1|0.8396|22.8568|0.2201|0.1904|0.8339|18.0344|87.1272|0.9270|0.9481|
> || GuardSplat-VU2|0.3765|6.4916|0.6594|0.1722|0.7374|56.8699|151.8966|0.6272|0.4956|
> || AdLift*-VU|0.6550|27.4526|0.3762|0.1791|0.8037|24.3330|102.6782|0.8337|0.8794|
>
>
> **Table D3:** Comparison with more baselines at the similar PSNR-level (training views). Bracketed values indicate changes relative to the unprotected baseline. Best results are highlighted in **bold**.
> | Scenes|Method|SSIM ($\uparrow$)|PSNR ($\uparrow$)|LPIPS ($\downarrow$)|CLIP_d($\downarrow$)|CLIP_s($\downarrow$)|FID ($\uparrow$)|F_1/8 ($\downarrow$)|F_8 ($\downarrow$)|
> |:-:|:-:|:-:|:-:|:-:|:-:|:-:|:-:|:-:|:-:|
> | Face|No Protection|0.9381|32.6436|0.0668|0.1685|0.8852|22.0270|0.9958|0.9947|
> ||GuardSplat-(SH)-VU|0.9348|31.7765|0.0802|0.1649 (-0.0036)|0.8769 (-0.0083)|26.2044 (+4.1774)|0.9952 (-0.0006)|0.9949 (0.0002)|
> ||GuardSplat-(SH)-VT|0.9319|28.8520|0.0930|0.1630 (-0.0055)|0.8643 (-0.0209)|28.4026 (+6.3756)|0.9923 (-0.0035)|0.9919 (-0.0028)|
> ||GuardSplat-(PC)-VU|0.8143|25.2876|0.1557|0.1605 (-0.0080)|0.8592 (-0.0260)|32.1277 (+10.1007)|0.9873 (-0.0085)|0.9910 (-0.0037)|
> ||GuardSplat-(PC)-VT|0.7979|24.4582|0.1779|0.1583 (-0.0102)|0.8456 (-0.0396)|37.3899 (+15.3629)|0.9813 (-0.0145)|0.9822 (-0.0125)|
> ||GuardSplat-(Full)-VU|0.8113|25.1306|0.1580|0.1617 (-0.0068)|0.8581 (-0.0271)|32.5037 (+10.4767)|0.9838 (-0.0120)|0.9887 (-0.0060)|
> ||GuardSplat-(Full)-VT|0.8016|24.6235|0.1775|0.1581 (-0.0104)|0.8493 (-0.0359)|36.9855 (+14.9585)|0.9816 (-0.0142)|0.9818 (-0.0129)|
> ||GaussianMarker-VU|0.9351|28.8343|0.0802|0.1596 (-0.0089)|0.8590 (-0.0262)|32.4431 (+10.4161)|0.9845 (-0.0113)|0.9916 (-0.0031)|
> ||GaussianMarker-VT|0.9364|28.8833|0.0720|0.1620 (-0.0065)|0.8665 (-0.0187)|26.3699 (+4.3429)|0.9929 (-0.0029)|0.9942 (-0.0005)|
> ||AdLift*-VU|0.7001|28.3963|0.3774|**0.1554 (-0.0131)**|**0.7620 (-0.1232)**|**88.5865 (+66.5595)**|**0.8954 (-0.1004)**|**0.8742 (-0.1205)**|
> | Fangzhou|No Protection|0.9218|32.1021|0.1093|0.1937|0.9080|23.7996|0.9817|0.9795|
> ||GuardSplat-(SH)-VU|0.8682|24.3325|0.1952|0.1882 (-0.0055)|0.8482 (-0.0598)|49.5017 (+25.7021)|0.9419 (-0.0398)|0.9716 (-0.0079)|
> ||GuardSplat-(SH)-VT|0.8727|24.2598|0.2198|0.1871 (-0.0066)|0.8588 (-0.0492)|43.0895 (+19.2899)|0.9325 (-0.0492)|0.9375 (-0.0420)|
> ||GuardSplat-(PC)-VU|0.8734|26.4272|0.1810|0.1863 (-0.0074)|0.8811 (-0.0269)|32.6075 (+8.8079)|0.9646 (-0.0171)|0.9669 (-0.0126)|
> ||GuardSplat-(PC)-VT|0.8683|27.0370|0.1756|0.1866 (-0.0071)|0.8804 (-0.0276)|35.9910 (+12.1914)|0.9138 (-0.0679)|0.9229 (-0.0566)|
> ||GuardSplat-(Full)-VU|0.8712|27.5267|0.1743|0.1906 (-0.0031)|0.8896 (-0.0184)|32.9438 (+9.1442)|0.9501 (-0.0316)|0.9547 (-0.0248)|
> ||GuardSplat-(Full)-VT|0.8590|27.0691|0.1821|0.1876 (-0.0061)|0.8778 (-0.0302)|36.4141 (+12.6145)|0.9598 (-0.0219)|0.9473 (-0.0322)|
> ||GaussianMarker-VU|0.9200|28.0355|0.1455|0.1907 (-0.0030)|0.8858 (-0.0222)|34.3784 (+10.5788)|0.9695 (-0.0122)|0.9672 (-0.0123)|
> ||GaussianMarker-VT|0.8637|23.7295|0.2460|0.1865 (-0.0072)|0.8296 (-0.0784)|57.1012 (+33.3016)|0.9015 (-0.0802)|0.9368 (-0.0427)|
> ||AdLift*-VU|0.6706|28.3497|0.3767|**0.1781 (-0.0156)**|**0.7900 (-0.1180)**|**84.2180 (+60.4184)**|**0.8030 (-0.1787)**|**0.7749 (-0.2046)**|

---

> ### Author Response · Authors · 2025-11-26
> **Response to Reviewer nZzV (Part 3/5)**
>
> **Table D4:** Comparison with more baselines at the similar PSNR-level (novel views). Bracketed values indicate changes relative to the unprotected baseline. Best results are highlighted in **bold**.
> | Scenes|Method|SSIM ($\uparrow$)|PSNR ($\uparrow$)|LPIPS ($\downarrow$)|CLIP_d($\downarrow$)|CLIP_s($\downarrow$)|FID ($\uparrow$)|F_1/8 ($\downarrow$)|F_8 ($\downarrow$)|
> |:-:|:-:|:-:|:-:|:-:|:-:|:-:|:-:|:-:|:-:|
> | Face|No Protection|0.8590|26.0390|0.1292|0.1704|0.8730|76.2210|0.9882|0.9835|
> ||GuardSplat-(SH)-VU|0.8552|25.7744|0.1442|0.1654 (-0.0050)|0.8666 (-0.0064)|83.8178 (+7.5968)|0.9834 (-0.0048)|0.9843 (0.0008)|
> ||GuardSplat-(SH)-VT|0.8528|25.0700|0.1571|0.1623 (-0.0081)|0.8550 (-0.0180)|80.3380 (+4.1170)|0.9901 (0.0019)|0.9879 (0.0044)|
> ||GuardSplat-(PC)-VU|0.7852|23.8763|0.1799|0.1667 (-0.0037)|0.8575 (-0.0155)|85.5826 (+9.3616)|0.9789 (-0.0093)|0.9790 (-0.0045)|
> ||GuardSplat-(PC)-VT|0.7712|23.2702|0.2004|0.1652 (-0.0052)|0.8480 (-0.0250)|91.1785 (+14.9575)|0.9752 (-0.0130)|0.9783 (-0.0052)|
> ||GuardSplat-(Full)-VU|0.7819|23.7877|0.1826|0.1658 (-0.0046)|0.8535 (-0.0195)|87.5892 (+11.3682)|0.9801 (-0.0081)|0.9762 (-0.0073)|
> ||GuardSplat-(Full)-VT|0.7752|23.4051|0.1983|0.1622 (-0.0082)|0.8505 (-0.0225)|90.4853 (+14.2643)|0.9748 (-0.0134)|0.9774 (-0.0061)|
> ||GaussianMarker-VU|0.8566|24.4911|0.1404|0.1628 (-0.0076)|0.8578 (-0.0152)|81.2042 (+4.9832)|0.9772 (-0.0110)|0.9770 (-0.0065)|
> ||GaussianMarker-VT|0.8567|24.8557|0.1346|0.1647 (-0.0057)|0.8608 (-0.0122)|81.1457 (+4.9247)|0.9849 (-0.0033)|0.9798 (-0.0037)|
> ||AdLift*-VU|0.6367|25.6115|0.4018|**0.1586 (-0.0118)**|**0.7786 (-0.0944)**|**129.8172 (+53.5962)**|**0.8807 (-0.1075)**|**0.9279 (-0.0556)**|
> | Fangzhou|No Protection|0.9010|30.2873|0.1161|0.1938|0.9053|51.7357|0.9905|0.9913|
> ||GuardSplat-(SH)-VU|0.8484|23.7358|0.2038|0.1905 (-0.0033)|0.8464 (-0.0589)|80.7992 (+29.0635)|0.9493 (-0.0412)|0.9679 (-0.0234)|
> ||GuardSplat-(SH)-VT|0.8546|23.6860|0.2254|0.1876 (-0.0062)|0.8570 (-0.0483)|71.4061 (+19.6704)|0.9533 (-0.0372)|0.9449 (-0.0464)|
> ||GuardSplat-(PC)-VU|0.8608|25.7609|0.1861|0.1880 (-0.0058)|0.8763 (-0.0290)|62.2475 (+10.5118)|0.9573 (-0.0332)|0.9629 (-0.0284)|
> ||GuardSplat-(PC)-VT|0.8570|26.3485|0.1807|0.1900 (-0.0038)|0.8807 (-0.0246)|65.9022 (+14.1665)|0.9403 (-0.0502)|0.9583 (-0.0330)|
> ||GuardSplat-(Full)-VU|0.8582|26.7445|0.1795|0.1903 (-0.0035)|0.8824 (-0.0229)|62.6535 (+10.9178)|0.9541 (-0.0364)|0.9543 (-0.0370)|
> ||GuardSplat-(Full)-VT|0.8480|26.3622|0.1866|0.1898 (-0.0040)|0.8752 (-0.0301)|67.569 (+15.8333)|0.9296 (-0.0609)|0.9403 (-0.0510)|
> ||GaussianMarker-VU|0.8998|26.8220|0.1516|0.1914 (-0.0024)|0.8844 (-0.0209)|64.0244 (+12.2887)|0.9735 (-0.0170)|0.9726 (-0.0187)|
> ||GaussianMarker-VT|0.8510|23.2438|0.2488|0.1872 (-0.0066)|0.8265 (-0.0788)|87.4162 (+35.6805)|0.8793 (-0.1112)|0.9319 (-0.0594)|
> ||AdLift*-VU|0.6550|27.4526|0.3762|**0.1791 (-0.0147)**|**0.8037 (-0.1016)**|**102.6782 (+50.9425)**|**0.8337 (-0.1568)**|**0.8794 (-0.1119)**|
>
>
>
> > **W2&Q2: Can the authors provide quantitative evidence of view consistency? For instance, how does AdLift compare to prior watermarking approaches like GaussianMarker or GuardSplat in terms of perturbation alignment across views?**
>
>
> **Response 2:** Thank you for the question. Since there is no ground-truth of 3DGS perturbation, directly quantifying cross-view perturbation consistency remains challenging. Existing methods such as GaussianMarker [D2] and GuardSplat [D1] similarly do not provide quantitative measures of perturbation alignment across viewpoints.
>
> **To approximate perturbation consistency across views, we compute the standard deviation of perceptual distortion metrics (SSIM, PSNR, LPIPS) across rendered views**, where lower values indicate more view-consistent perturbations. We evaluate both the original watermarking version of GaussianMarker and its anti-editing adaptation, and compare them with AdLift.
>
> From the results in **Table D5**, the perturbations learned by AdLift exhibit a comparable standard deviation in perceptual distortion metrics relative to existing methods. This indicates that lifting adversarial perturbations into the 3DGS space does not introduce additional view inconsistency compared with existing watermarking approaches.
>
> **Table D5:** Standard deviation of perceptual distortion metrics (lower is better).
> | Method|Train SSIM-std|Train PSNR-std|Train LPIPS-std|Novel SSIM-std|Novel PSNR-std|Novel LPIPS-std|
> |:-:|:-:|:-:|:-:|:-:|:-:|:-:|
> | GaussianMarker (for watermarking)|0.0189|1.4110|0.0208|0.0374|2.1331|0.0258|
> | GaussianMarker (for anti-editing)|**0.0172**|1.8121|**0.0178**|**0.0345**|1.9429|0.0241|
> | AdLift-VU (for anti-editing)|0.0191|**0.5225**|0.0246|0.0398|**1.4081**|**0.0193**|

---

> ### Author Response · Authors · 2025-11-26
> **Response to Reviewer nZzV (Part 4/5)**
>
> > **W3&Q3: Does duplicating the Gaussians lead to significant memory overhead, particularly for dense scenes where the number of primitives is already high? A discussion or analysis of the scalability implications would be helpful.**
>
> **Response 3:** Thank you for pointing this out. To quantify the memory and computational overhead of Adlift, we report runtime, memory usage, and model size in **Table D6**, using the Face scene as a representative example. It is important to note that the duplicated Gaussians are not permanently retained. **During optimization, AdLift applies the standard densify-and-prune strategy, which means that Gaussian primitives are only temporarily duplicated to improve representational capacity and are later pruned when unnecessary.** Thus, the final model size grows moderately.
>
> Besides, the increased memory and time cost is mainly due to the need to load and back-propagate through the editing model across multiple rendered views to achieve view-generalizable protection. The computation scales with both the editing model complexity and the number of supervised viewpoints.
>
> This reflects an inherent challenge of adversarial learning for 3DGS and highlights opportunities for future efficiency improvements, such as using lightweight surrogate editing models or adaptive view-sampling strategies. We have added this discussion to the Limitations section in the revised version.
>
> **Table D6:** Computational resources for the Face scene, measured on a single RTX 4090 GPU (24 GB). The original resolution for Face scene (994 × 738) is used.
> | Stage|Time/Iteration|Total Time|Memory (Training)|#Gaussians|
> |:-:|:-:|:-:|:-:|:-:|
> | Pretrain: 3DGS|~0.02s|~9min (30k iteration)|~3GB|906,867|
> | Continue: AdLift-VU|~0.8s|~45min (5k iteration)|~19GB|1,309,732|
>
>
>
>
> > **Q4: Can the authors report additional evaluation metrics such as FID or Precision, in addition to CLIP score? Prior 2D image protection works like PhotoGuard and AdvDM typically report FID and Precision, which are widely adopted and allow for better comparability. It would also be helpful to provide these metrics for the editing results of the baseline methods discussed in Q1.**
>
> **Response 4:** Thank you for the helpful suggestion. Following your suggestion, we have incorporated additional evaluation metrics from [D5-D6] to complement the existing CLIP-based evaluation:
> - **Additional metrics.** we additionally report the FID score [D5] and the PRD score [D6] to more comprehensively assess anti-editing performance. Specifically,
>   - **FID** measures how close generated images are to real ones by comparing their Gaussian statistics in the Inception feature space using the Fréchet distance. Lower scores indicate higher similarity, thus a *higher FID score corresponds to stronger resistance to editing*.
>   - **PRD** reports two standard summary values (**F$_{1/8}$** and **F$_8$**) reflecting precision–recall trade-offs across generative distributions.
>     - F$_{1/8}$: A precision-dominant F-measure, emphasizing sample quality (realism and fidelity) over diversity.
>     - F$_{8}$: A recall-dominant F-measure, emphasizing diversity and distributional coverage over precision.
>     - Lower F$_{1/8}$ and F$_8$ values indicate stronger anti-editing performance.
> - **Implementation details.** we follow the evaluation protocol used in [D3], where metrics are computed by measuring the similarity between edits generated from protected and non-protected images.
> - **Results.** For response to **Q1**, we provide these metrics for our method and all baseline, with detailed results shown in **Table D1-D4**. Furthermore, in all additional experiments during rebuttal, we also include these three metrics (FID, F$_{1/8}$, and F$_8$) alongside the original CLIP score. Across all metrics, our method consistently outperforms existing baselines, demonstrating that the improvement is not limited to semantic alignment (CLIP scores) but also holds under more distribution-oriented measures such as FID and PRD.

---

> ### Author Response · Authors · 2025-11-26
> **Response to Reviewer nZzV (Part 5/5)**
>
> > **Q5: In Figure 8, the qualitative results of local 3DGS editing (e.g., the bear scene) show only marginal visual differences between the unprotected and protected assets. In contrast, Table 2 reports that the proposed AdLift-ST achieves substantially stronger protection performance (e.g., lower CLIPd) compared with Instruct-GS2GS and DGE (Global). How does AdLift-ST yield such strong quantitative gains despite the visual outputs in Figure 8 appearing relatively similar on the bear scene? Additional explanation would help clarify the effectiveness of local protection.**
>
> **Response 5:** Thank you for the thoughtful question. We argue that the discrepancy arises because **CLIP$_d$ measures the degree of semantic alignment with the editing instruction**. Therefore, **for local editing**, its variation depends on whether CLIP can detect **fine-grained semantic changes before and after editing**. As such, in some cases, even if the visual difference appears appears subtle to the human eye, the protection may already suppress key semantic attributes required for the editing instruction. For example, in the Face scene (**Figure 8** in the main paper), the edit output of protected 3DGS lacks features such as exaggerated makeup compared to the edit output of unprotected 3DGS, and in the Bear scene, distinctive grizzly or polar bear characteristics are suppressed in the protected version. As a result, CLIP$_d$ decreases substantially, reflecting that the editing instruction is no longer being semantically followed.
>
> In contrast, **for 2D global and 3D global editing**, even when protection visibly degrades the edit quality, the edited outputs may still follow the instruction semantics. For instance, in the bronze bear case (**Figure 7** in the main paper), although the protected output is noticeably distorted, features such as a bronze or golden bear head remain, which CLIP may still interprets as satisfying the instruction prompt. Similarly, in **Figure 6**, under the instruction “Turn him into an old man with wrinkles”, the protected low-quality edit still contains semantic cues such as wrinkles and old man, resulting in only a moderate reduction in CLIP$_d$.
>
> ---
>
> [D1] GuardSplat: Efficient and Robust Watermarking for 3D Gaussian Splatting. CVPR 2025.\
> [D2] GaussianMarker: Uncertainty-Aware Copyright Protection of 3D Gaussian Splatting. NeurIPS 2024.\
> [D3] Raising the cost of malicious ai-powered image editing. ICML 2023.\
> [D4] Adversarial Example Does Good: Preventing Painting Imitation from Diffusion Models via Adversarial Examples. ICML 2023.\
> [D5] GANs Trained by a Two Time-Scale Update Rule Converge to a Local Nash Equilibrium. NIPS 2017.\
> [D6] Assessing Generative Models via Precision and Recall. NIPS 2018.

---

### Official Review · Reviewer_WspE · 2025-10-26

**Soundness:** 2
**Presentation:** 3
**Contribution:** 2
**Rating:** 4
**Confidence:** 3

**Summary:**

This paper proposes an adversarial safeguard for 3D Gaussian Splatting that protects against unauthorized instruction-driven editing. It lifts bounded 2D adversarial perturbations into 3D as safeguard Gaussians, optimized via a Lifted Projected Gradient Descent that alternates between gradient truncation and image-to-Gaussian fitting. This design ensures view-generalizable and imperceptible protection across both training and novel viewpoints.

**Strengths:**

- The paper addresses an interesting and practical problem in anti-editing for 3D content protection.

- The proposed method is technically sound and reasonable.

- Experimental results across multiple scenes demonstrate performance improvements.

**Weaknesses:**

- The technical innovation appears limited. The proposed approach essentially adds adversarial perturbations to 2D images and then reconstructs them via Gaussian Splatting. While effective, it largely adopts conventional 2D anti-editing optimization with an additional 3DGS reconstruction step, which reduces its novelty and technical contribution to the community.

- The experimental evaluation is limited to only four scenes, which may be insufficient to robustly assess the generality and effectiveness of the method.

- The paper lacks analysis on transferability — it remains unclear how well the optimized perturbations perform against other editing networks beyond the one used during training.

- The robustness of the perturbations to purification or denoising processes is not discussed, leaving uncertainty about their effectiveness under potential real-world defenses.

**Questions:**

Please refer to the Weakness part above.

---

> ### Author Response · Authors · 2025-11-26
> **Response to Reviewer WspE (Part 1/5)**
>
> Dear Reviewer `WspE`,
>
> Thanks for your valuable comments! We address the weaknesses below. Please kindly let us know if you have further concerns.
>
> > **W1: The technical innovation appears limited. The proposed approach essentially adds adversarial perturbations to 2D images and then reconstructs them via Gaussian Splatting. While effective, it largely adopts conventional 2D anti-editing optimization with an additional 3DGS reconstruction step, which reduces its novelty and technical contribution to the community.**
>
>
> **Response 1:** Thank you for the comment, and we apologize for any confusion. We here clarify that our contribution is not merely applying 2D adversarial optimization followed by 3D reconstruction, but addressing the practical challenge of enabling anti-editing protection in 3D Gaussian Splatting, where perturbations must remain view-consistent, perceptually bounded at rendering, and robust across 3D-aware editing pipelines. More importantly, unlike 2D pixel-based images, 3DGS stores information in Gaussian primitives with heterogeneous attributes, making direct transfer from 2D methods non-trivial. Specifically, this setting introduces several core challenges:
> - **View inconsistency of 2D adversarial perturbations:**
>   - 2D adversarial perturbations are view-inconsistent. Applying 2D adversarial optimization (like [C1-C2]) followed by 3D reconstruction may induces cross-view conflicts, and the resulting 3DGS suffers from underfitting and exhibits poor generalization to novel views. Our results in **Figure 2** and **Figure 9** in the main paper offer evidence supporting this point.
> - **Soft constraint limitations:**
>   - Although prior 3DGS watermarking methods (e.g., [C3-C4]) introduce imperceptible perturbations using soft regularization on Gaussian parameters, directly applying 2D anti-editing objectives through such mechanisms fails to achieve a proper balance between invisibility and adversarial effectiveness. Our experiments in **Tables 6-9** and **Figures 3, 10, 11, 28 and 29** in the revised manuscript offers evidence supporting this point.
> - **No unified perturbation space:**
>   - 3DGS parameters consist of heterogeneous attributes (position, scale, opacity, SH coefficients), making it non-trivial to impose a unified perturbation budget analogous to PGD in 2D pixel space.
>
> To address these issues and enable anti-editing for 3DGS, we therefore introduce AdLift, a framework designed to lift PGD-like adversarial perturbations to the 3D Gaussian space. **Technically, our contribution includes:**
> - A PGD-style optimization framework for 3DGS that enforces strict perceptual bounds at the rendering level during iterative optimization.
> - A two-stage update strategy: (i) gradient truncation in rendering space to satisfy adversarial constraints, followed by (ii) image-to-Gaussian fitting to propagate perturbations back to 3D Gaussian parameters.
>
> This design enables the first demonstration of hard-constrained adversarial perturbations for 3DGS editing protection, achieving view-consistent and editing-resilient behavior across multiple editing models.
>
> We have clarified these points and refined the wording in the revised manuscript to prevent misunderstanding.
>
>
>
>
> > **W2: The experimental evaluation is limited to only four scenes, which may be insufficient to robustly assess the generality and effectiveness of the method.**
>
> **Response 2:** Thank you for the suggestion. We have extended our evaluation by including two additional 360-degree scenes. The corresponding quantitative results are reported in **Table C1** (training views) and **Table C2** (novel views). Besides, qualitative results are shown in **Figure 34** in the revised manuscript. Across all metrics and scenes, our method consistently outperforms both unprotected 3DGS assets and baseline methods in terms of anti-editing effectiveness.

---

> ### Author Response · Authors · 2025-11-26
> **Response to Reviewer WspE (Part 2/5)**
>
> **Table C1:** Performance of AdLift on more scenes (training views). Bracketed values indicate changes relative to the unprotected baseline. Best results are highlighted in **bold**.
> | Scene|Method|SSIM ($\uparrow$)|PSNR ($\uparrow$)|LPIPS ($\downarrow$)|CLIP_d ($\downarrow$)|CLIP_s ($\downarrow$)|FID ($\uparrow$)|F_1/8 ($\downarrow$)|F_8 ($\downarrow$)|
> |:-:|:-:|:-:|:-:|:-:|:-:|:-:|:-:|:-:|:-:|
> | Bicycle|No Protection|0.9132|27.5901|0.0536|0.1935|0.8995|17.1704|0.9947|0.9944|
> ||Fit2D-VU|0.8084|25.5982|0.1818|0.1914 (-0.0021)|0.8714 (-0.0281)|25.6811 (+8.5107)|0.9740 (-0.0207)|0.9860 (-0.0084)|
> ||Fit2D-VT|0.8186|25.8145|0.1126|0.1889 (-0.0046)|0.8805 (-0.0190)|23.3785 (+6.2081)|0.9863 (-0.0084)|0.9885 (-0.0059)|
> ||AdLift*-VU|0.7822|25.8218|0.2388|0.1851 (-0.0084)|**0.8250 (-0.0745)**|**48.7802 (+31.6098)**|**0.9218 (-0.0729)**|**0.9600 (-0.0344)**|
> ||AdLift*-VT|0.7899|26.0401|0.1259|**0.1837 (-0.0098)**|0.8589 (-0.0406)|30.5979 (+13.4275)|0.9714 (-0.0233)|0.9825 (-0.0119)|
> | Stump|No Protection|0.9456|32.2618|0.0353|0.1689|0.9298|26.0890|0.9954|0.9964|
> ||Fit2D-VU|0.7976|27.2747|0.2089|**0.1651 (-0.0038)**|0.8846 (-0.0452)|47.1604 (+21.0714)|0.9716 (-0.0238)|0.9739 (-0.0225)|
> ||Fit2D-VT|0.8145|28.0254|0.1172|0.1676 (-0.0013)|0.8953 (-0.0345)|42.4515 (+16.3625)|0.9493 (-0.0461)|0.9666 (-0.0298)|
> ||AdLift*-VU|0.7838|27.4652|0.2616|0.1657 (-0.0032)|**0.8491 (-0.0807)**|**66.9380 (+40.8490)**|0.8910 (-0.1044)|0.9021 (-0.0943)|
> ||AdLift*-VT|0.8001|28.2688|0.1358|0.1669 (-0.0020)|0.8739 (-0.0559)|59.8454 (+33.7564)|**0.8851 (-0.1103)**|**0.8864 (-0.1100)**|
>
>
> **Table C2:** Performance of AdLift on more scenes (novel views). Bracketed values indicate changes relative to the unprotected baseline. Best results are highlighted in **bold**.
> | Scene|Method|SSIM ($\uparrow$)|PSNR ($\uparrow$)|LPIPS ($\downarrow$)|CLIP_d ($\downarrow$)|CLIP_s ($\downarrow$)|FID ($\uparrow$)|F_1/8 ($\downarrow$)|F_8 ($\downarrow$)|
> |:-:|:-:|:-:|:-:|:-:|:-:|:-:|:-:|:-:|:-:|
> | Bicycle|No Protection|0.8431|26.743|0.0672|0.1914|0.9019|53.7759|0.9826|0.9840|
> ||Fit2D-VU|0.7306|24.9763|0.2004|0.1897 (-0.0017)|0.1874 (-0.7145)|61.5747 (+7.7988)|0.9659 (-0.0167)|0.9725 (-0.0115)|
> ||Fit2D-VT|0.7446|25.3495|0.1297|0.1848 (-0.0066)|0.8839 (-0.0180)|61.2288 (+7.4529)|0.9671 (-0.0155)|0.9743 (-0.0097)|
> ||AdLift*-VU|0.7019|24.4851|0.2650|0.1844 (-0.0070)|**0.8384 (-0.0635)**|**91.9488 (+38.1729)**|**0.8726 (-0.1100)**|**0.9275 (-0.0565)**|
> ||AdLift*-VT|0.7152|24.7930|0.1533|**0.1798 (-0.0116)**|0.8583 (-0.0436)|76.4528 (+22.6769)|0.9367 (-0.0459)|0.9564 (-0.0276)|
> | Stump|No Protection|0.8157|27.2135|0.0713|0.1652|0.9223|82.9066|0.9826|0.9852|
> ||Fit2D-VU|0.6754|24.5672|0.2095|**0.1593 (-0.0059)**|0.8933 (-0.0290)|103.8607 (+20.9541)|0.9702 (-0.0124)|0.9545 (-0.0307)|
> ||Fit2D-VT|0.6971|25.1350|0.1403|0.1607 (-0.0045)|0.9042 (-0.0181)|95.7885 (+12.8819)|0.9584 (-0.0242)|0.9732 (-0.0120)|
> ||AdLift*-VU|0.6607|24.3507|0.2498|0.1602 (-0.0050)|**0.8685 (-0.0538)**|**114.4635 (+31.5569)**|0.9023 (-0.0803)|**0.9174 (-0.0678)**|
> ||AdLift*-VT|0.6884|25.0868|0.1599|0.1602 (-0.0050)|0.8882 (-0.0341)|104.7286 (+21.8220)|**0.8873 (-0.0953)**|0.9181 (-0.0671)|

---

> ### Author Response · Authors · 2025-11-26
> **Response to Reviewer WspE (Part 3/5)**
>
> > **W3: The paper lacks analysis on transferability — it remains unclear how well the optimized perturbations perform against other editing networks beyond the one used during training.**
>
>
> **Response 3:** Thank you for the valuable comment. We additionally evaluate whether AdLift, which aims at lifting 2D adversarial perturbations into the 3D Gaussian space, can inherits the transferability in 2D adversarial protection.
>
> **Experimental setup:** We train AdLift using SD-v1.5-IP2P [C5] as the surrogate editing model (the same as our paper), and then test it against three unseen editing models, including different editing pipelines and fine-tuned variants:
> - MagicBrush [C6]: Enhanced fine-tuned version of SDv1.5-IP2P on MagicBrush dataset.
> - SDEdit [C7]: A diffusion-based editing framework that generates edited images by noising and denoising an input image, without requiring instruction-conditioned finetuning.
> - SDXL-IP2P [C5]: Instruction fine-tuning of Stable Diffusion XL (SDXL).
>
> **Results:** The quantitative results of transferability are reported in **Table C3** (training views) and **Table C4** (novel views). Qualitative results are shown in **Figure 32** in the revised manuscript. Empirically, across all unseen editing models, AdLift can degrade editing qualities compared to the unprotected 3DGS asset, demonstrating that the protective effect generalizes beyond the surrogate model. This suggests that the proposed AdLift preserves the transferability behavior of 2D adversarial perturbations, even after being lifted into the 3D Gaussian domain.
>
> **Table C3:** Transferability of AdLift (training views). Bracketed values indicate changes relative to the unprotected baseline. Best results are highlighted in **bold**.
> | Scenes|Edit Pipelines|Method|CLIP_d ($\downarrow$)|CLIP_s ($\downarrow$)|FID ($\uparrow$)|F_1/8 ($\downarrow$)|F_8 ($\downarrow$)|
> |:-:|:-:|:-:|:-:|:-:|:-:|:-:|:-:|
> | Face|SDv1.5-IP2P|No Protection|0.1685|0.8852|22.0270|0.9958|0.9947|
> |||AdLift*-VU|**0.1554 (-0.0131)**|**0.7620 (-0.1232)**|**88.5865 (+66.5595)**|**0.8954 (-0.1004)**|**0.8742 (-0.1205)**|
> |||AdLift*-VT|0.1561 (-0.0124)|0.8272 (-0.0580)|45.1351 (+23.1081)|0.9545 (-0.0413)|0.9650 (-0.0297)|
> ||MagicBrush|No Protection|0.1741|0.8776|22.1594|0.9926|0.9885|
> |||AdLift*-VU|**0.1420 (-0.0321)**|**0.7457 (-0.1319)**|**69.4154 (+47.2560)**|**0.9305 (-0.0621)**|**0.9297 (-0.0588)**|
> |||AdLift*-VT|0.1588 (-0.0153)|0.8138 (-0.0638)|48.8487 (+26.6893)|0.9642 (-0.0284)|0.9540 (-0.0345)|
> ||SDEdit|No Protection|0.1659|0.7811|25.2357|0.9913|0.9894|
> |||AdLift*-VU|**0.1448 (-0.0211)**|0.7141 (-0.067)|58.5985 (+33.3628)|0.8436 (-0.1477)|**0.9484 (-0.0410)**|
> |||AdLift*-VT|0.1633 (-0.0026)|**0.7131 (-0.068)**|**62.6101 (+37.3744)**|**0.8372 (-0.1541)**|0.9532 (-0.0362)|
> ||SDXL-IP2P|No Protection|0.1053|0.7124|26.7935|0.9908|0.9899|
> |||AdLift*-VU|0.1031 (-0.0022)|**0.6509 (-0.0615)**|**48.6225 (+21.8290)**|**0.9691 (-0.0217)**|0.9540 (-0.0359)|
> |||AdLift*-VT|**0.1016 (-0.0037)**|0.6681 (-0.0443)|44.9257 (+18.1322)|0.9771 (-0.0137)|**0.9512 (-0.0387)**|
> | Fangzhou|SDv1.5-IP2P|No Protection|0.1937|0.9080|23.7996|0.9817|0.9795|
> |||AdLift*-VU|0.1781 (-0.0156)|**0.7900 (-0.1180)**|**84.2180 (+60.4184)**|**0.8030 (-0.1787)**|**0.7749 (-0.2046)**|
> |||AdLift*-VT|**0.1776 (-0.0161)**|0.8198 (-0.0882)|56.5018 (+32.7022)|0.8721 (-0.1096)|0.8839 (-0.0956)|
> ||MagicBrush|No Protection|0.1839|0.8583|40.1794|0.8484|0.7938|
> |||AdLift*-VU|0.1625 (-0.0214)|**0.7548 (-0.1035)**|**105.8215 (+65.6421)**|0.7135 (-0.1349)|0.5795 (-0.2143)|
> |||AdLift*-VT|**0.1611 (-0.0228)**|0.8021 (-0.0562)|89.8512 (+49.6718)|**0.5955 (-0.2529)**|**0.5720 (-0.2218)**|
> ||SDEdit|No Protection|0.1224|0.7587|19.3086|0.9785|0.9795|
> |||AdLift*-VU|0.1213 (-0.0011)|0.7148 (-0.0439)|88.8757 (+69.5671)|0.6992 (-0.2793)|0.8442 (-0.1353)|
> |||AdLift*-VT|**0.1179 (-0.0045)**|**0.6794 (-0.0793)**|**109.4655 (+90.1569)**|**0.5921 (-0.3864)**|**0.7453 (-0.2342)**|
> ||SDXL-IP2P|No Protection|0.0777|0.7533|29.4324|0.9422|0.9106|
> |||AdLift*-VU|**0.0719 (-0.0058)**|**0.7322 (-0.0211)**|**47.7706 (+18.3382)**|0.9354 (-0.0068)|0.8700 (-0.0406)|
> |||AdLift*-VT|0.0738 (-0.0039)|0.7338 (-0.0195)|44.6247 (+15.1923)|**0.9273 (-0.0149)**|**0.8019 (-0.1087)**|

---

> ### Author Response · Authors · 2025-11-26
> **Response to Reviewer WspE (Part 4/5)**
>
> **Table C4:** Transferability of AdLift (novel views). Bracketed values indicate changes relative to the unprotected baseline. Best results are highlighted in **bold**.
> | Scenes|Edit Pipelines|Method|CLIP_d ($\downarrow$)|CLIP_s ($\downarrow$)|FID ($\uparrow$)|F_1/8 ($\downarrow$)|F_8 ($\downarrow$)|
> |:-:|:-:|:-:|:-:|:-:|:-:|:-:|:-:|
> | Face|IP2P|No Protection|0.1704|0.8730|76.2210|0.9882|0.9835|
> |||AdLift*-VU|**0.1586 (-0.0118)**|**0.7786 (-0.0944)**|**129.8172 (+53.5962)**|**0.8807 (-0.1075)**|**0.9279 (-0.0556)**|
> |||AdLift*-VT|0.1595 (-0.0109)|0.8285 (-0.0445)|93.4765 (+17.2555)|0.9473 (-0.0409)|0.9694 (-0.0141)|
> ||MagicBrush|No Protection|0.1769|0.8635|78.1133|0.9784|0.9638|
> |||AdLift*-VU|**0.1475 (-0.0294)**|**0.7659 (-0.0976)**|**120.0269 (+41.9136)**|**0.9150 (-0.0634)**|**0.9145 (-0.0493)**|
> |||AdLift*-VT|0.1607 (-0.0162)|0.8148 (-0.0487)|100.0880 (+21.9747)|0.9657 (-0.0127)|0.9570 (-0.0068)|
> ||SDEdit|No Protection|0.1708|0.7727|89.2094|0.9568|0.9722|
> |||AdLift*-VU|**0.1554 (-0.0154)**|0.7397 (-0.033)|104.9600 (+15.7506)|0.9618 (0.0050)|0.9575 (-0.0147)|
> |||AdLift*-VT|0.1751 (0.0043)|**0.7353 (-0.0374)**|**118.0674 (+28.8580)**|**0.8806 (-0.0762)**|**0.9443 (-0.0279)**|
> ||SDXL-IP2P|No Protection|0.1140|0.7012|93.6249|0.9707|0.9672|
> |||AdLift*-VU|0.1094 (-0.0046)|**0.6481 (-0.0531)**|**115.9670 (+22.3421)**|**0.9214 (-0.0493)**|0.9289 (-0.0383)|
> |||AdLift*-VT|**0.1081 (-0.0059)**|0.6693 (-0.0319)|112.9657 (+19.3408)|0.9383 (-0.0324)|**0.9274 (-0.0398)**|
> | Fangzhou|IP2P|No Protection|0.1938|0.9053|51.7357|0.9905|0.9913|
> |||AdLift*-VU|0.1791 (-0.0147)|**0.8037 (-0.1016)**|**102.6782 (+50.9425)**|**0.8337 (-0.1568)**|0.8794 (-0.1119)|
> |||AdLift*-VT|**0.1788 (-0.0150)**|0.8148 (-0.0905)|86.0002 (+34.2645)|0.8862 (-0.1043)|**0.8676 (-0.1237)**|
> ||MagicBrush|No Protection|0.1812|0.8566|73.9738|0.9032|0.8124|
> |||AdLift*-VU|0.1605 (-0.0207)|**0.7608 (-0.0958)**|**144.9615 (+70.9877)**|0.7101 (-0.1931)|0.6113 (-0.2011)|
> |||AdLift*-VT|**0.1581 (-0.0231)**|0.7998 (-0.0568)|124.9021 (+50.9283)|**0.6838 (-0.2194)**|**0.5643 (-0.2481)**|
> ||SDEdit|No Protection|0.1220|0.7578|68.4325|0.9802|0.9621|
> |||AdLift*-VU|0.1292 (0.0072)|0.7300 (-0.0278)|124.4493 (+56.0168)|0.8783 (-0.1019)|0.9191 (-0.0430)|
> |||AdLift*-VT|**0.1194 (-0.0026)**|**0.6755 (-0.0823)**|**165.0488 (+96.6163)**|**0.6478 (-0.3324)**|**0.6729 (-0.2892)**|
> ||SDXL-IP2P|No Protection|0.0738|0.7496|75.9508|0.9628|0.9113|
> |||AdLift*-VU|**0.0695 (-0.0043)**|**0.7321 (-0.0175)**|**100.9508 (+25.0000)**|**0.9127 (-0.0501)**|0.9112 (-0.0001)|
> |||AdLift*-VT|0.0715 (-0.0023)|0.7331 (-0.0165)|93.3964 (+17.4456)|0.9346 (-0.0282)|**0.8424 (-0.0689)**|
>
>
> > **W4: The robustness of the perturbations to purification or denoising processes is not discussed, leaving uncertainty about their effectiveness under potential real-world defenses.**
>
> **Response 4:** Thank you for the suggestion. To evaluate robustness under purification, we follow the evaluation protocol used in prior 2D anti-editing works [C1–C2] to test our method against two purification strategies, including a diffusion-based purification method (DiffPure [C8]) and a standard compression method (JPEG compression [C9]). The quantitative results are reported in **Table C5** (training views) and **Table C6** (novel views). Qualitative results are shown in **Figure 33** in the revised manuscript.
>
> Our findings show that AdLift exhibit similar behavior to 2D adversarial protection: **AdLift still remains effective under JPEG compression-based purification and generalize across unseen views. While diffusion-based purification weakens the protection, AdLift still outperforms unprotected 3DGS assets.**
>
> We have added this discussion to the Limitations section, and we consider improving purification robustness, potentially by lifting purification-resilient 2D adversarial strategies into the 3DGS domain, as an important direction for future work.
>
> **Table C5:** Robustness of AdLift against purification methods (training Views). Bracketed values indicate changes relative to the unprotected baseline.
> | Scenes|Method|CLIP_d($\downarrow$)|CLIP_s($\downarrow$)|FID ($\uparrow$)|F_1/8 ($\downarrow$)|F_8 ($\downarrow$)|
> |:-:|:-:|:-:|:-:|:-:|:-:|:-:|
> | Face|No Protection|0.1685|0.8852|22.0270|0.9958|0.9947|
> ||AdLift*-VU|0.1554 (-0.0131)|0.7620 (-0.1232)|88.5865 (+66.5595)|0.8954 (-0.1004)|0.8742 (-0.1205)|
> ||AdLift*-VU-JPEG|0.1548 (-0.0137)|0.7851 (-0.1001)|69.2879 (+47.2609)|0.9375 (-0.0583)|0.9359 (-0.0588)|
> ||AdLift*-VU-Diffpure|0.1549 (-0.0136)|0.8173 (-0.0679)|51.5645 (+29.5375)|0.9753 (-0.0205)|0.9727 (-0.0220)|
> | Fangzhou|No Protection|0.1937|0.9080|23.7996|0.9817|0.9795|
> ||AdLift*-VU|0.1781 (-0.0156)|0.7900 (-0.1180)|84.2180 (+60.4184)|0.8030 (-0.1787)|0.7749 (-0.2046)|
> ||AdLift*-VU-JPEG|0.1791 (-0.0146)|0.8218 (-0.0862)|61.7341 (+37.9345)|0.9197 (-0.0620)|0.9332 (-0.0463)|
> ||AdLift*-VU-Diffpure|0.1860 (-0.0077)|0.8640 (-0.0440)|35.8747 (+12.0751)|0.9777 (-0.0040)|0.9670 (-0.0125)|

---

> ### Author Response · Authors · 2025-11-26
> **Response to Reviewer WspE (Part 5/5)**
>
> **Table C6:** Robustness of AdLift against purification methods (novel Views). Bracketed values indicate changes relative to the unprotected baseline.
> | Scenes|Method|CLIP_d($\downarrow$)|CLIP_s($\downarrow$)|FID ($\uparrow$)|F_1/8 ($\downarrow$)|F_8 ($\downarrow$)|
> |:-:|:-:|:-:|:-:|:-:|:-:|:-:|
> | Face|No Protection|0.1704|0.8730|76.2210|0.9882|0.9835|
> ||AdLift*-VU|0.1586 (-0.0118)|0.7786 (-0.0944)|129.8172 (+53.5962)|0.8807 (-0.1075)|0.9279 (-0.0556)|
> ||AdLift*-VU-JPEG|0.1573 (-0.0131)|0.7947 (-0.0783)|116.7051 (+40.4841)|0.9420 (-0.0462)|0.9486 (-0.0349)|
> ||AdLift*-VU-Diffpure|0.1573 (-0.0131)|0.8162 (-0.0568)|107.5986 (+31.3776)|0.9639 (-0.0243)|0.9432 (-0.0403)|
> | Fangzhou|No Protection|0.1938|0.9053|51.7357|0.9905|0.9913|
> ||AdLift*-VU|0.1791 (-0.0147)|0.8037 (-0.1016)|102.6782 (+50.9425)|0.8337 (-0.1568)|0.8794 (-0.1119)|
> ||AdLift*-VU-JPEG|0.1822 (-0.0116)|0.8229 (-0.0824)|85.8403 (+34.1046)|0.9353 (-0.0552)|0.9519 (-0.0394)|
> ||AdLift*-VU-Diffpure|0.1883 (-0.0055)|0.8618 (-0.0435)|64.7646 (+13.0289)|0.9756 (-0.0149)|0.9687 (-0.0226)|
>
> ---
>
> [C1] Adversarial Example Does Good: Preventing Painting Imitation from Diffusion Models via Adversarial Examples. ICML 2023.\
> [C2] Towards Effective Protection Against Diffusion-based Mimicry Through Score Distillation. ICLR 2024.\
> [C3] GuardSplat: Efficient and Robust Watermarking for 3D Gaussian Splatting. CVPR 2025.\
> [C4] GaussianMarker: Uncertainty-Aware Copyright Protection of 3D Gaussian Splatting. NeurIPS 2024.\
> [C5] InstructPix2Pix: Learning To Follow Image Editing Instructions. CVPR 2023.\
> [C6] MagicBrush: A Manually Annotated Dataset for Instruction-Guided Image Editing. NeurIPS 2023.\
> [C7] SDEdit: Guided Image Synthesis and Editing with Stochastic Differential Equations. ICLR 2022.\
> [C8] Diffusion Models for Adversarial Purification. ICML 2022.\
> [C9] JPEG Compressed Images Can Bypass Protections Against AI Editing. Arxiv 2023.

---

### Official Review · Reviewer_NAMb · 2025-10-27

**Soundness:** 3
**Presentation:** 3
**Contribution:** 2
**Rating:** 4
**Confidence:** 4

**Summary:**

This paper proposes AdLift, a framework that lifts bounded 2D adversarial perturbations into 3D Gaussian Splatting (3DGS) representations to defend against unauthorized instruction-driven editing. The method introduces a tailored Lifted PGD that alternates between gradient truncation and image-to-Gaussian fitting, aiming to balance invisibility and protection strength. Experiments across multiple 3DGS scenes demonstrate improved protection against both 2D and 3D instruction-based editing, while maintaining reasonable visual fidelity

**Strengths:**

1. The paper highlights an important and timely issue, security risks of instruction-driven 3DGS editing, making it a potentially impactful line of research.

2. The proposed Lifted PGD framework is presented with mathematical detail, algorithmic steps, and visual diagrams, which aids clarity.

3. Evidence of feasibility: The experiments, though limited, demonstrate that adversarial perturbations can be adapted from 2D to 3DGS to some extent, offering preliminary validation.

**Weaknesses:**

1. The core idea is essentially extending 2D PGD adversarial training into 3D Gaussian Splatting, with rendering-space constraints for imperceptibility. This is more of an adaptation than a fundamentally new paradigm. The paper repeatedly claims to be the first to propose active protection for 3DGS, but prior work on watermarking, Gaussian perturbations, and 2D adversarial protection already covers similar ground. The novelty is overstated. Theoretical contribution is thin: the method description relies heavily on equations and algorithmic flowcharts, without rigorous analysis or formal guarantees of convergence.

2. The method assumes white-box access to editing models, which is unrealistic in practice. No experiments or discussion are provided for black-box or transfer scenarios.Lifted PGD requires repeated rendering and optimization, which could be computationally expensive. No runtime, memory, or scalability analysis is reported.

3. Baselines are limited to Fit2D and GuardSplat, which are relatively weak. Stronger or more recent defenses are missing, raising concerns that the comparisons are intentionally favorable. Only a narrow set of editing models are tested (IP2P, Instruct-GS2GS, DGE). The method’s robustness against more advanced or diverse editing pipelines is not demonstrated.

4. Quantitative improvements are modest, and in some cases, invisibility metrics (SSIM, PSNR, LPIPS) actually degrade, undermining the “balance” argument. Visual examples (e.g., Fig. 6–8) show protected assets yielding “unnatural” or distorted edits. This could easily be interpreted as a byproduct of noise rather than a principled defense.

**Questions:**

I will raise my score if the authors address W1 and W2.

---

> ### Author Response · Authors · 2025-11-26
> **Response to Reviewer NAMb (Part 1/5)**
>
> Dear Reviewer `NAMb`,
>
> Thanks for your valuable comments! We address the weaknesses below. Please kindly let us know if you have further concerns.
>
> > **W1: The core idea is essentially extending 2D PGD adversarial training into 3D Gaussian Splatting, with rendering-space constraints for imperceptibility. This is more of an adaptation than a fundamentally new paradigm. The paper repeatedly claims to be the first to propose active protection for 3DGS, but prior work on watermarking, Gaussian perturbations, and 2D adversarial protection already covers similar ground. The novelty is overstated. Theoretical contribution is thin: the method description relies heavily on equations and algorithmic flowcharts, without rigorous analysis or formal guarantees of convergence.**
>
> **Response 1:** Thank you for the valuable feedback, and we apologize for the confusion. We have refined and clarified our statements in the revised manuscript and added discussion on how our work relates and differs to prior research directions. Specifically, we refine our statement to:
> - **Research problem:** We explore anti-editing protection in the context of 3D Gaussian Splatting.
> - **Methodology:** We adapt PGD-style, hard-constrained adversarial perturbations directly in 3DGS space to achieve view-consistent protection.
>
> Although related prior directions exist, they differ in goals and applicability:
> - **3DGS watermarking** [B1-B2]
>   - **Problem scope:** 3DGS watermarking only provides passive traceability rather than actively resisting instruction-driven editing.
>   - **Methodology:** Existing 3DGS watermarking approaches, which inject perturbations into Gaussian parameters relying on soft invisibility regularization over all (or selected) attributes, fail to balance between invisibility and adversarial effectiveness when applied to learning imperceptible adversarial perturbations against editing models. Our results from the rebuttal period (**Table B4–B5**), together with the experiments in the main paper (**Figures 3, 10, 11, 28 and 29**), provide evidence supporting this point.
> - **2D adversarial protection** [B3-B4]
>   - Although the goal is also anti-editing, these methods cannot be directly applied to 3DGS. As 3DGS representations use 3D Gaussian primitives instead of pixel-space representations. Specifically, 3D Gaussian parameters are heterogeneous, and different attributes encode distinct physical meanings (e.g., position, scale, orientation, appearance, and opacity), making it non-trivial to impose a unified perturbation budget analogous to PGD in 2D pixel space.
>
> Taken together, **prior directions inform but do not solve our problem setting**, which requires adversarially optimized, perceptually bounded, and view-consistent protection for 3DGS under instruction-driven editing pipelines.
>
> Regarding theoretical analysis, we acknowledge that the current work is primarily empirical. Providing formal convergence guarantees for adversarial optimization in the context of 3D Gaussian Splatting is non-trivial due to the heterogeneous parameterization and the complexity of the differentiable rendering process. We view this as an important open research direction rather than a solved component of the field, and we have added a discussion in the Limitations section to clarify the intended scope of this work.
>
>
>
> >**W2.1: The method assumes white-box access to editing models, which is unrealistic in practice. No experiments or discussion are provided for black-box or transfer scenarios.**
>
> **Response 2.1:** Thank you for the valuable comment. We additionally evaluate whether AdLift, which aims at lifting 2D adversarial perturbations into the 3D Gaussian space, can inherits the transferability in 2D adversarial protection.
>
> **Experimental setup:** We train AdLift using SD-v1.5-IP2P [B5] as the surrogate editing model (the same as our paper), and then test it against three unseen editing models, including different editing pipelines and fine-tuned variants:
> - MagicBrush [B6]: Enhanced fine-tuned version of SDv1.5-IP2P on MagicBrush dataset.
> - SDEdit [B7]: A diffusion-based editing framework that generates edited images by noising and denoising an input image, without requiring instruction-conditioned finetuning.
> - SDXL-IP2P [B5]: Instruction fine-tuning of Stable Diffusion XL (SDXL).
>
> **Results:** The quantitative results of transferability are reported in **Table B1** (training views) and **Table B2** (novel views). Qualitative results are shown in **Figure 32** in the revised manuscript. Empirically, across all unseen editing models, AdLift can degrade editing qualities compared to the unprotected 3DGS asset, demonstrating that the protective effect generalizes beyond the surrogate model. This suggests that the proposed AdLift preserves the transferability behavior of 2D adversarial perturbations, even after being lifted into the 3D Gaussian domain.

---

> ### Author Response · Authors · 2025-11-26
> **Response to Reviewer NAMb (Part 2/5)**
>
> **Table B1:** Transferability of AdLift (training views). Bracketed values indicate changes relative to the unprotected baseline. Best results are highlighted in **bold**.
> | Scenes|Edit Pipelines|Method|CLIP_d ($\downarrow$)|CLIP_s ($\downarrow$)|FID ($\uparrow$)|F_1/8 ($\downarrow$)|F_8 ($\downarrow$)|
> |:-:|:-:|:-:|:-:|:-:|:-:|:-:|:-:|
> | Face|SDv1.5-IP2P|No Protection|0.1685|0.8852|22.0270|0.9958|0.9947|
> |||AdLift*-VU|**0.1554 (-0.0131)**|**0.7620 (-0.1232)**|**88.5865 (+66.5595)**|**0.8954 (-0.1004)**|**0.8742 (-0.1205)**|
> |||AdLift*-VT|0.1561 (-0.0124)|0.8272 (-0.0580)|45.1351 (+23.1081)|0.9545 (-0.0413)|0.9650 (-0.0297)|
> ||MagicBrush|No Protection|0.1741|0.8776|22.1594|0.9926|0.9885|
> |||AdLift*-VU|**0.1420 (-0.0321)**|**0.7457 (-0.1319)**|**69.4154 (+47.2560)**|**0.9305 (-0.0621)**|**0.9297 (-0.0588)**|
> |||AdLift*-VT|0.1588 (-0.0153)|0.8138 (-0.0638)|48.8487 (+26.6893)|0.9642 (-0.0284)|0.9540 (-0.0345)|
> ||SDEdit|No Protection|0.1659|0.7811|25.2357|0.9913|0.9894|
> |||AdLift*-VU|**0.1448 (-0.0211)**|0.7141 (-0.067)|58.5985 (+33.3628)|0.8436 (-0.1477)|**0.9484 (-0.0410)**|
> |||AdLift*-VT|0.1633 (-0.0026)|**0.7131 (-0.068)**|**62.6101 (+37.3744)**|**0.8372 (-0.1541)**|0.9532 (-0.0362)|
> ||SDXL-IP2P|No Protection|0.1053|0.7124|26.7935|0.9908|0.9899|
> |||AdLift*-VU|0.1031 (-0.0022)|**0.6509 (-0.0615)**|**48.6225 (+21.8290)**|**0.9691 (-0.0217)**|0.9540 (-0.0359)|
> |||AdLift*-VT|**0.1016 (-0.0037)**|0.6681 (-0.0443)|44.9257 (+18.1322)|0.9771 (-0.0137)|**0.9512 (-0.0387)**|
> | Fangzhou|SDv1.5-IP2P|No Protection|0.1937|0.9080|23.7996|0.9817|0.9795|
> |||AdLift*-VU|0.1781 (-0.0156)|**0.7900 (-0.1180)**|**84.2180 (+60.4184)**|**0.8030 (-0.1787)**|**0.7749 (-0.2046)**|
> |||AdLift*-VT|**0.1776 (-0.0161)**|0.8198 (-0.0882)|56.5018 (+32.7022)|0.8721 (-0.1096)|0.8839 (-0.0956)|
> ||MagicBrush|No Protection|0.1839|0.8583|40.1794|0.8484|0.7938|
> |||AdLift*-VU|0.1625 (-0.0214)|**0.7548 (-0.1035)**|**105.8215 (+65.6421)**|0.7135 (-0.1349)|0.5795 (-0.2143)|
> |||AdLift*-VT|**0.1611 (-0.0228)**|0.8021 (-0.0562)|89.8512 (+49.6718)|**0.5955 (-0.2529)**|**0.5720 (-0.2218)**|
> ||SDEdit|No Protection|0.1224|0.7587|19.3086|0.9785|0.9795|
> |||AdLift*-VU|0.1213 (-0.0011)|0.7148 (-0.0439)|88.8757 (+69.5671)|0.6992 (-0.2793)|0.8442 (-0.1353)|
> |||AdLift*-VT|**0.1179 (-0.0045)**|**0.6794 (-0.0793)**|**109.4655 (+90.1569)**|**0.5921 (-0.3864)**|**0.7453 (-0.2342)**|
> ||SDXL-IP2P|No Protection|0.0777|0.7533|29.4324|0.9422|0.9106|
> |||AdLift*-VU|**0.0719 (-0.0058)**|**0.7322 (-0.0211)**|**47.7706 (+18.3382)**|0.9354 (-0.0068)|0.8700 (-0.0406)|
> |||AdLift*-VT|0.0738 (-0.0039)|0.7338 (-0.0195)|44.6247 (+15.1923)|**0.9273 (-0.0149)**|**0.8019 (-0.1087)**|

---

> ### Author Response · Authors · 2025-11-26
> **Response to Reviewer NAMb (Part 3/5)**
>
> **Table B2:** Transferability of AdLift (novel views). Bracketed values indicate changes relative to the unprotected baseline. Best results are highlighted in **bold**.
> | Scenes|Edit Pipelines|Method|CLIP_d ($\downarrow$)|CLIP_s ($\downarrow$)|FID ($\uparrow$)|F_1/8 ($\downarrow$)|F_8 ($\downarrow$)|
> |:-:|:-:|:-:|:-:|:-:|:-:|:-:|:-:|
> | Face|SDv1.5-IP2P|No Protection|0.1704|0.8730|76.2210|0.9882|0.9835|
> |||AdLift*-VU|**0.1586 (-0.0118)**|**0.7786 (-0.0944)**|**129.8172 (+53.5962)**|**0.8807 (-0.1075)**|**0.9279 (-0.0556)**|
> |||AdLift*-VT|0.1595 (-0.0109)|0.8285 (-0.0445)|93.4765 (+17.2555)|0.9473 (-0.0409)|0.9694 (-0.0141)|
> ||MagicBrush|No Protection|0.1769|0.8635|78.1133|0.9784|0.9638|
> |||AdLift*-VU|**0.1475 (-0.0294)**|**0.7659 (-0.0976)**|**120.0269 (+41.9136)**|**0.9150 (-0.0634)**|**0.9145 (-0.0493)**|
> |||AdLift*-VT|0.1607 (-0.0162)|0.8148 (-0.0487)|100.0880 (+21.9747)|0.9657 (-0.0127)|0.9570 (-0.0068)|
> ||SDEdit|No Protection|0.1708|0.7727|89.2094|0.9568|0.9722|
> |||AdLift*-VU|**0.1554 (-0.0154)**|0.7397 (-0.033)|104.9600 (+15.7506)|0.9618 (0.0050)|0.9575 (-0.0147)|
> |||AdLift*-VT|0.1751 (0.0043)|**0.7353 (-0.0374)**|**118.0674 (+28.8580)**|**0.8806 (-0.0762)**|**0.9443 (-0.0279)**|
> ||SDXL-IP2P|No Protection|0.1140|0.7012|93.6249|0.9707|0.9672|
> |||AdLift*-VU|0.1094 (-0.0046)|**0.6481 (-0.0531)**|**115.9670 (+22.3421)**|**0.9214 (-0.0493)**|0.9289 (-0.0383)|
> |||AdLift*-VT|**0.1081 (-0.0059)**|0.6693 (-0.0319)|112.9657 (+19.3408)|0.9383 (-0.0324)|**0.9274 (-0.0398)**|
> | Fangzhou|SDv1.5-IP2P|No Protection|0.1938|0.9053|51.7357|0.9905|0.9913|
> |||AdLift*-VU|0.1791 (-0.0147)|**0.8037 (-0.1016)**|**102.6782 (+50.9425)**|**0.8337 (-0.1568)**|0.8794 (-0.1119)|
> |||AdLift*-VT|**0.1788 (-0.0150)**|0.8148 (-0.0905)|86.0002 (+34.2645)|0.8862 (-0.1043)|**0.8676 (-0.1237)**|
> ||MagicBrush|No Protection|0.1812|0.8566|73.9738|0.9032|0.8124|
> |||AdLift*-VU|0.1605 (-0.0207)|**0.7608 (-0.0958)**|**144.9615 (+70.9877)**|0.7101 (-0.1931)|0.6113 (-0.2011)|
> |||AdLift*-VT|**0.1581 (-0.0231)**|0.7998 (-0.0568)|124.9021 (+50.9283)|**0.6838 (-0.2194)**|**0.5643 (-0.2481)**|
> ||SDEdit|No Protection|0.1220|0.7578|68.4325|0.9802|0.9621|
> |||AdLift*-VU|0.1292 (0.0072)|0.7300 (-0.0278)|124.4493 (+56.0168)|0.8783 (-0.1019)|0.9191 (-0.0430)|
> |||AdLift*-VT|**0.1194 (-0.0026)**|**0.6755 (-0.0823)**|**165.0488 (+96.6163)**|**0.6478 (-0.3324)**|**0.6729 (-0.2892)**|
> ||SDXL-IP2P|No Protection|0.0738|0.7496|75.9508|0.9628|0.9113|
> |||AdLift*-VU|**0.0695 (-0.0043)**|**0.7321 (-0.0175)**|**100.9508 (+25.0000)**|**0.9127 (-0.0501)**|0.9112 (-0.0001)|
> |||AdLift*-VT|0.0715 (-0.0023)|0.7331 (-0.0165)|93.3964 (+17.4456)|0.9346 (-0.0282)|**0.8424 (-0.0689)**|
>
>
>
> > **W2.2: Lifted PGD requires repeated rendering and optimization, which could be computationally expensive. No runtime, memory, or scalability analysis is reported.**
>
> **Response 2.2:** Thank you for pointing this out. We agree that lifting PGD into the 3DGS space introduces additional computational overhead. To quantify this, we report runtime, memory usage, and model size in **Table B3**, using the Face scene as a representative example. The increased cost is mainly due to the need to back-propagate gradients through the editing model on multiple rendered views in order to achieve view-generalizable protection. The computation scales with both the editing model complexity and the number of supervised viewpoints.
>
> This reflects an inherent challenge of adversarial learning for 3DGS and highlights opportunities for future efficiency improvements, such as using lightweight surrogate editing models or adaptive view-sampling strategies. We have added this discussion to the Limitations section in the revised version.
>
> **Table B3:** Computational resources for the Face scene, measured on a single RTX 4090 GPU (24 GB). The original resolution for Face scene (994 × 738) is used.
> | Stage|Time/Iteration|Total Time|Memory (Training)|#Gaussians|
> |:-:|:-:|:-:|:-:|:-:|
> | Pretrain: 3DGS|~0.02s|~9min (30k iteration)|~3GB|906,867|
> | Continue: AdLift-VU|~0.8s|~45min (5k iteration)|~19GB|1,309,732|

---

> ### Author Response · Authors · 2025-11-26
> **Response to Reviewer NAMb (Part 4/5)**
>
> > **W3: Baselines are limited to Fit2D and GuardSplat, which are relatively weak. The method’s robustness against more advanced or diverse editing pipelines is not demonstrated.**
>
> **Response 3:** Thank you for your suggestion. Regarding robustness against additional editing pipelines, we have included new results in **Response 2.1**, where AdLift is evaluated under three different and unseen editing pipelines. **In addition, to provide a broader comparison, we include several potential baselines by adapting representative 3DGS watermarking and parameter-perturbation methods into an adversarial training regime, and compare them against AdLift under matched perceptual quality.** Specifically, we include the following variants:
> - GuardSplat-(SH): Perturb only the *Spherical-Harmonic* (SH) features of all Gaussians, following the original GuardSplat [B1] design.
> - GuardSplat-(PC): Perturb *Positions* and *Covariance* features of all Gaussians using GuardSplat [B1].
> - GuardSplat-(Full): Perturb all features of all Gaussians using GuardSplat [B1], including *Position*, *Covariance*, *Opacity*, and *SH features*.
> - GaussianMarker [B2]: Split low-uncertainty Gaussians and optimize the newly added Gaussians to encode perturbations.
>
> For each method, we evaluate two adversarial objectives, untargeted VAE loss (VU) and targeted VAE loss (VT). We preserve each method’s original fidelity-preserving loss but replace original watermark decoding loss with our adversarial objective so that all methods are optimized toward the same protection goal.
>
> **Results:** The quantitative results are provided in **Table B4** (training views) and **Table B5** (novel views). We report protection performance under similar perceptual quality (similar PSNR). We also include representative qualitative comparisons in **Figure 30** (training views) and **Figure 31** (novel views) in the revised manuscript.  From the results, AdLift consistently achieves stronger anti-editing performance under similar invisibility constraints.
>
> **Table B4:** Comparison with more baselines at the similar PSNR-level (training views). Bracketed values indicate changes relative to the unprotected baseline. Best results are highlighted in **bold**.
> | Scenes|Method|SSIM ($\uparrow$)|PSNR ($\uparrow$)|LPIPS ($\downarrow$)|CLIP_d($\downarrow$)|CLIP_s($\downarrow$)|FID ($\uparrow$)|F_1/8 ($\downarrow$)|F_8 ($\downarrow$)|
> |:-:|:-:|:-:|:-:|:-:|:-:|:-:|:-:|:-:|:-:|
> | Face|No Protection|0.9381|32.6436|0.0668|0.1685|0.8852|22.0270|0.9958|0.9947|
> ||GuardSplat-(SH)-VU|0.9348|31.7765|0.0802|0.1649 (-0.0036)|0.8769 (-0.0083)|26.2044 (+4.1774)|0.9952 (-0.0006)|0.9949 (0.0002)|
> ||GuardSplat-(SH)-VT|0.9319|28.8520|0.0930|0.1630 (-0.0055)|0.8643 (-0.0209)|28.4026 (+6.3756)|0.9923 (-0.0035)|0.9919 (-0.0028)|
> ||GuardSplat-(PC)-VU|0.8143|25.2876|0.1557|0.1605 (-0.0080)|0.8592 (-0.0260)|32.1277 (+10.1007)|0.9873 (-0.0085)|0.9910 (-0.0037)|
> ||GuardSplat-(PC)-VT|0.7979|24.4582|0.1779|0.1583 (-0.0102)|0.8456 (-0.0396)|37.3899 (+15.3629)|0.9813 (-0.0145)|0.9822 (-0.0125)|
> ||GuardSplat-(Full)-VU|0.8113|25.1306|0.1580|0.1617 (-0.0068)|0.8581 (-0.0271)|32.5037 (+10.4767)|0.9838 (-0.0120)|0.9887 (-0.0060)|
> ||GuardSplat-(Full)-VT|0.8016|24.6235|0.1775|0.1581 (-0.0104)|0.8493 (-0.0359)|36.9855 (+14.9585)|0.9816 (-0.0142)|0.9818 (-0.0129)|
> ||GaussianMarker-VU|0.9351|28.8343|0.0802|0.1596 (-0.0089)|0.8590 (-0.0262)|32.4431 (+10.4161)|0.9845 (-0.0113)|0.9916 (-0.0031)|
> ||GaussianMarker-VT|0.9364|28.8833|0.0720|0.1620 (-0.0065)|0.8665 (-0.0187)|26.3699 (+4.3429)|0.9929 (-0.0029)|0.9942 (-0.0005)|
> ||AdLift*-VU|0.7001|28.3963|0.3774|**0.1554 (-0.0131)**|**0.7620 (-0.1232)**|**88.5865 (+66.5595)**|**0.8954 (-0.1004)**|**0.8742 (-0.1205)**|
> | Fangzhou|No Protection|0.9218|32.1021|0.1093|0.1937|0.9080|23.7996|0.9817|0.9795|
> ||GuardSplat-(SH)-VU|0.8682|24.3325|0.1952|0.1882 (-0.0055)|0.8482 (-0.0598)|49.5017 (+25.7021)|0.9419 (-0.0398)|0.9716 (-0.0079)|
> ||GuardSplat-(SH)-VT|0.8727|24.2598|0.2198|0.1871 (-0.0066)|0.8588 (-0.0492)|43.0895 (+19.2899)|0.9325 (-0.0492)|0.9375 (-0.0420)|
> ||GuardSplat-(PC)-VU|0.8734|26.4272|0.1810|0.1863 (-0.0074)|0.8811 (-0.0269)|32.6075 (+8.8079)|0.9646 (-0.0171)|0.9669 (-0.0126)|
> ||GuardSplat-(PC)-VT|0.8683|27.0370|0.1756|0.1866 (-0.0071)|0.8804 (-0.0276)|35.9910 (+12.1914)|0.9138 (-0.0679)|0.9229 (-0.0566)|
> ||GuardSplat-(Full)-VU|0.8712|27.5267|0.1743|0.1906 (-0.0031)|0.8896 (-0.0184)|32.9438 (+9.1442)|0.9501 (-0.0316)|0.9547 (-0.0248)|
> ||GuardSplat-(Full)-VT|0.8590|27.0691|0.1821|0.1876 (-0.0061)|0.8778 (-0.0302)|36.4141 (+12.6145)|0.9598 (-0.0219)|0.9473 (-0.0322)|
> ||GaussianMarker-VU|0.9200|28.0355|0.1455|0.1907 (-0.0030)|0.8858 (-0.0222)|34.3784 (+10.5788)|0.9695 (-0.0122)|0.9672 (-0.0123)|
> ||GaussianMarker-VT|0.8637|23.7295|0.2460|0.1865 (-0.0072)|0.8296 (-0.0784)|57.1012 (+33.3016)|0.9015 (-0.0802)|0.9368 (-0.0427)|
> ||AdLift*-VU|0.6706|28.3497|0.3767|**0.1781 (-0.0156)**|**0.7900 (-0.1180)**|**84.2180 (+60.4184)**|**0.8030 (-0.1787)**|**0.7749 (-0.2046)**|

---

> ### Author Response · Authors · 2025-11-26
> **Response to Reviewer NAMb (Part 5/5)**
>
> **Table B5:** Comparison with more baselines at the similar PSNR-level (novel views). Bracketed values indicate changes relative to the unprotected baseline. Best results are highlighted in **bold**.
> | Scenes|Method|SSIM ($\uparrow$)|PSNR ($\uparrow$)|LPIPS ($\downarrow$)|CLIP_d($\downarrow$)|CLIP_s($\downarrow$)|FID ($\uparrow$)|F_1/8 ($\downarrow$)|F_8 ($\downarrow$)|
> |:-:|:-:|:-:|:-:|:-:|:-:|:-:|:-:|:-:|:-:|
> | Face|No Protection|0.8590|26.0390|0.1292|0.1704|0.8730|76.2210|0.9882|0.9835|
> ||GuardSplat-(SH)-VU|0.8552|25.7744|0.1442|0.1654 (-0.0050)|0.8666 (-0.0064)|83.8178 (+7.5968)|0.9834 (-0.0048)|0.9843 (0.0008)|
> ||GuardSplat-(SH)-VT|0.8528|25.0700|0.1571|0.1623 (-0.0081)|0.8550 (-0.0180)|80.3380 (+4.1170)|0.9901 (0.0019)|0.9879 (0.0044)|
> ||GuardSplat-(PC)-VU|0.7852|23.8763|0.1799|0.1667 (-0.0037)|0.8575 (-0.0155)|85.5826 (+9.3616)|0.9789 (-0.0093)|0.9790 (-0.0045)|
> ||GuardSplat-(PC)-VT|0.7712|23.2702|0.2004|0.1652 (-0.0052)|0.8480 (-0.0250)|91.1785 (+14.9575)|0.9752 (-0.0130)|0.9783 (-0.0052)|
> ||GuardSplat-(Full)-VU|0.7819|23.7877|0.1826|0.1658 (-0.0046)|0.8535 (-0.0195)|87.5892 (+11.3682)|0.9801 (-0.0081)|0.9762 (-0.0073)|
> ||GuardSplat-(Full)-VT|0.7752|23.4051|0.1983|0.1622 (-0.0082)|0.8505 (-0.0225)|90.4853 (+14.2643)|0.9748 (-0.0134)|0.9774 (-0.0061)|
> ||GaussianMarker-VU|0.8566|24.4911|0.1404|0.1628 (-0.0076)|0.8578 (-0.0152)|81.2042 (+4.9832)|0.9772 (-0.0110)|0.9770 (-0.0065)|
> ||GaussianMarker-VT|0.8567|24.8557|0.1346|0.1647 (-0.0057)|0.8608 (-0.0122)|81.1457 (+4.9247)|0.9849 (-0.0033)|0.9798 (-0.0037)|
> ||AdLift*-VU|0.6367|25.6115|0.4018|**0.1586 (-0.0118)**|**0.7786 (-0.0944)**|**129.8172 (+53.5962)**|**0.8807 (-0.1075)**|**0.9279 (-0.0556)**|
> | Fangzhou|No Protection|0.9010|30.2873|0.1161|0.1938|0.9053|51.7357|0.9905|0.9913|
> ||GuardSplat-(SH)-VU|0.8484|23.7358|0.2038|0.1905 (-0.0033)|0.8464 (-0.0589)|80.7992 (+29.0635)|0.9493 (-0.0412)|0.9679 (-0.0234)|
> ||GuardSplat-(SH)-VT|0.8546|23.6860|0.2254|0.1876 (-0.0062)|0.8570 (-0.0483)|71.4061 (+19.6704)|0.9533 (-0.0372)|0.9449 (-0.0464)|
> ||GuardSplat-(PC)-VU|0.8608|25.7609|0.1861|0.1880 (-0.0058)|0.8763 (-0.0290)|62.2475 (+10.5118)|0.9573 (-0.0332)|0.9629 (-0.0284)|
> ||GuardSplat-(PC)-VT|0.8570|26.3485|0.1807|0.1900 (-0.0038)|0.8807 (-0.0246)|65.9022 (+14.1665)|0.9403 (-0.0502)|0.9583 (-0.0330)|
> ||GuardSplat-(Full)-VU|0.8582|26.7445|0.1795|0.1903 (-0.0035)|0.8824 (-0.0229)|62.6535 (+10.9178)|0.9541 (-0.0364)|0.9543 (-0.0370)|
> ||GuardSplat-(Full)-VT|0.8480|26.3622|0.1866|0.1898 (-0.0040)|0.8752 (-0.0301)|67.569 (+15.8333)|0.9296 (-0.0609)|0.9403 (-0.0510)|
> ||GaussianMarker-VU|0.8998|26.8220|0.1516|0.1914 (-0.0024)|0.8844 (-0.0209)|64.0244 (+12.2887)|0.9735 (-0.0170)|0.9726 (-0.0187)|
> ||GaussianMarker-VT|0.8510|23.2438|0.2488|0.1872 (-0.0066)|0.8265 (-0.0788)|87.4162 (+35.6805)|0.8793 (-0.1112)|0.9319 (-0.0594)|
> ||AdLift*-VU|0.6550|27.4526|0.3762|**0.1791 (-0.0147)**|**0.8037 (-0.1016)**|**102.6782 (+50.9425)**|**0.8337 (-0.1568)**|**0.8794 (-0.1119)**|
>
>
> > **W4: Quantitative improvements are modest, and in some cases, invisibility metrics (SSIM, PSNR, LPIPS) actually degrade, undermining the “balance” argument. Visual examples (e.g., Fig. 6–8) show protected assets yielding “unnatural” or distorted edits. This could easily be interpreted as a byproduct of noise rather than a principled defense.**
>
>
> **Response 4:** We are sorry for the confusion. Our use of the term **balance** refers to the observation that **AdLift consistently achieves stronger editing resistance while maintaining similar or better perceptual quality compared to a broader baselines.** Our extensive experiments in **Table B4–B5**, together with the experiments in the main paper (**Figures 3, 10, 11, 28 and 29**), provide evidence supporting this point. This also indicates that the learned perturbations are not trivial noise.
>
> To avoid confusion, we have added additional clarification in the revised manuscript. In addition, we agree that further reducing perceptual impact while retaining protection strength is an important direction, and we have added this point to the Limitation section.
>
>
> ---
>
> [B1] GuardSplat: Efficient and Robust Watermarking for 3D Gaussian Splatting. CVPR 2025.\
> [B2] GaussianMarker: Uncertainty-Aware Copyright Protection of 3D Gaussian Splatting. NeurIPS 2024.\
> [B3] Adversarial Example Does Good: Preventing Painting Imitation from Diffusion Models via Adversarial Examples. ICML 2023.\
> [B4] Towards Effective Protection Against Diffusion-based Mimicry Through Score Distillation. ICLR 2024.\
> [B5] InstructPix2Pix: Learning To Follow Image Editing Instructions. CVPR 2023.\
> [B6] MagicBrush: A Manually Annotated Dataset for Instruction-Guided Image Editing. NeurIPS 2023.\
> [B7] SDEdit: Guided Image Synthesis and Editing with Stochastic Differential Equations. ICLR 2022.

---

> > ### Comment · Reviewer_NAMb · 2025-11-27
> >
> > Thanks for the effort. I have raised my score since the authors addressed most of my concerns.

---

> > > ### Author Response · Authors · 2025-11-27
> > >
> > > Thank you very much for your time and the positive update. We truly appreciate your constructive feedback throughout the review process, and we are glad to hear that our clarifications addressed your concerns. Thank you again for your consideration.

---

### Official Review · Reviewer_ebSz · 2025-10-30

**Soundness:** 3
**Presentation:** 3
**Contribution:** 3
**Rating:** 4
**Confidence:** 2

**Summary:**

The paper introduces AdLift, a novel framework to actively protect 3D Gaussian Splatting assets from unauthorized instruction-based editing. It leverages adversarial perturbations lifted from the 2D rendered image domain into 3D Gaussian space using a custom Lifted Projected Gradient Descent algorithm. The method enforces strict rendering-space bounds to ensure visual fidelity while maintaining strong protection against 2D and 3D editing pipelines such as InstructPix2Pix and Instruct-GS2GS. Experiments on several 3D scenes show that AdLift achieves strong protection with minimal perceptual distortion.

**Strengths:**

1. The idea of lifting perturbations from 2D to 3D to achieve view-consistent adversarial protection is novel and well-motivated.
2. The paper is well-written and well-organized
3. Comprehensive experiments across 2D image editing, global, and local 3DGS editing show consistent results.

**Weaknesses:**

1. The proposed method assumes full white-box access to the editing model (e.g., InstructPix2Pix or Instruct-GS2GS) to obtain gradients for adversarial optimization. However, in practice, the attacker may use a variety of proprietary or black-box diffusion-based models. The perturbations optimized against one editing model may not generalize to others. The paper lacks experiments to show the transferability and robustness of the method effectiveness across models, a “protected” 3DGS asset might still be editable or corrupted when attacked with a different model or pipeline.
2. The paper only compares against GuardSplat and Fit2D, omitting broader baselines such as adversarially trained 3D representations, or certified perturbation approaches.
3. There is no detailed ablation study analyzing the sensitivity of the method to key hyperparameters.

**Questions:**

See above

---

> ### Author Response · Authors · 2025-11-25
> **Response to Reviewer ebSz (Part 1/4)**
>
> Dear Reviewer `ebSz`,
>
> Thanks for your valuable reviews! We address your concerns as follows. Please let us know if anything remains unclear.
>
> > **W1: The paper lacks experiments to show the transferability and robustness of the method effectiveness across models, a “protected” 3DGS asset might still be editable or corrupted when attacked with a different model or pipeline.**
>
>
> **Response 1:** Thank you for the valuable comment. We additionally evaluate whether AdLift, which aims at lifting 2D adversarial perturbations into the 3D Gaussian space, can inherits the transferability in 2D adversarial protection.
>
> **Experimental setup:** We train AdLift using SD-v1.5-IP2P [A1] as the surrogate editing model (the same as our paper), and then test it against three unseen editing models, including different editing pipelines and fine-tuned variants:
> - MagicBrush [A2]: Enhanced fine-tuned version of SDv1.5-IP2P on MagicBrush dataset.
> - SDEdit [A3]: A diffusion-based editing framework that generates edited images by noising and denoising an input image, without requiring instruction-conditioned fine-tuning.
> - SDXL-IP2P [A1]: Instruction fine-tuning of Stable Diffusion XL (SDXL).
>
> **Results:** The quantitative results of transferability are reported in **Table A1** (training views) and **Table A2** (novel views). Qualitative results are shown in **Figure 32** in the revised manuscript. Empirically, across all unseen editing models, AdLift can degrade editing qualities compared to the unprotected 3DGS asset, demonstrating that the protective effect generalizes beyond the surrogate model. This suggests that the proposed AdLift preserves the transferability behavior of 2D adversarial perturbations, even after being lifted into the 3D Gaussian domain.
>
>
>
> **Table A1:** Transferability of AdLift (training views). Bracketed values indicate changes relative to the unprotected baseline. Best results are highlighted in **bold**.
> | Scenes|Edit Pipelines|Method|CLIP_d ($\downarrow$)|CLIP_s ($\downarrow$)|FID ($\uparrow$)|F_1/8 ($\downarrow$)|F_8 ($\downarrow$)|
> |:-:|:-:|:-:|:-:|:-:|:-:|:-:|:-:|
> | Face|SDv1.5-IP2P|No Protection|0.1685|0.8852|22.0270|0.9958|0.9947|
> |||AdLift*-VU|**0.1554 (-0.0131)**|**0.7620 (-0.1232)**|**88.5865 (+66.5595)**|**0.8954 (-0.1004)**|**0.8742 (-0.1205)**|
> |||AdLift*-VT|0.1561 (-0.0124)|0.8272 (-0.0580)|45.1351 (+23.1081)|0.9545 (-0.0413)|0.9650 (-0.0297)|
> ||MagicBrush|No Protection|0.1741|0.8776|22.1594|0.9926|0.9885|
> |||AdLift*-VU|**0.1420 (-0.0321)**|**0.7457 (-0.1319)**|**69.4154 (+47.2560)**|**0.9305 (-0.0621)**|**0.9297 (-0.0588)**|
> |||AdLift*-VT|0.1588 (-0.0153)|0.8138 (-0.0638)|48.8487 (+26.6893)|0.9642 (-0.0284)|0.9540 (-0.0345)|
> ||SDEdit|No Protection|0.1659|0.7811|25.2357|0.9913|0.9894|
> |||AdLift*-VU|**0.1448 (-0.0211)**|0.7141 (-0.067)|58.5985 (+33.3628)|0.8436 (-0.1477)|**0.9484 (-0.0410)**|
> |||AdLift*-VT|0.1633 (-0.0026)|**0.7131 (-0.068)**|**62.6101 (+37.3744)**|**0.8372 (-0.1541)**|0.9532 (-0.0362)|
> ||SDXL-IP2P|No Protection|0.1053|0.7124|26.7935|0.9908|0.9899|
> |||AdLift*-VU|0.1031 (-0.0022)|**0.6509 (-0.0615)**|**48.6225 (+21.8290)**|**0.9691 (-0.0217)**|0.9540 (-0.0359)|
> |||AdLift*-VT|**0.1016 (-0.0037)**|0.6681 (-0.0443)|44.9257 (+18.1322)|0.9771 (-0.0137)|**0.9512 (-0.0387)**|
> | Fangzhou|SDv1.5-IP2P|No Protection|0.1937|0.9080|23.7996|0.9817|0.9795|
> |||AdLift*-VU|0.1781 (-0.0156)|**0.7900 (-0.1180)**|**84.2180 (+60.4184)**|**0.8030 (-0.1787)**|**0.7749 (-0.2046)**|
> |||AdLift*-VT|**0.1776 (-0.0161)**|0.8198 (-0.0882)|56.5018 (+32.7022)|0.8721 (-0.1096)|0.8839 (-0.0956)|
> ||MagicBrush|No Protection|0.1839|0.8583|40.1794|0.8484|0.7938|
> |||AdLift*-VU|0.1625 (-0.0214)|**0.7548 (-0.1035)**|**105.8215 (+65.6421)**|0.7135 (-0.1349)|0.5795 (-0.2143)|
> |||AdLift*-VT|**0.1611 (-0.0228)**|0.8021 (-0.0562)|89.8512 (+49.6718)|**0.5955 (-0.2529)**|**0.5720 (-0.2218)**|
> ||SDEdit|No Protection|0.1224|0.7587|19.3086|0.9785|0.9795|
> |||AdLift*-VU|0.1213 (-0.0011)|0.7148 (-0.0439)|88.8757 (+69.5671)|0.6992 (-0.2793)|0.8442 (-0.1353)|
> |||AdLift*-VT|**0.1179 (-0.0045)**|**0.6794 (-0.0793)**|**109.4655 (+90.1569)**|**0.5921 (-0.3864)**|**0.7453 (-0.2342)**|
> ||SDXL-IP2P|No Protection|0.0777|0.7533|29.4324|0.9422|0.9106|
> |||AdLift*-VU|**0.0719 (-0.0058)**|**0.7322 (-0.0211)**|**47.7706 (+18.3382)**|0.9354 (-0.0068)|0.8700 (-0.0406)|
> |||AdLift*-VT|0.0738 (-0.0039)|0.7338 (-0.0195)|44.6247 (+15.1923)|**0.9273 (-0.0149)**|**0.8019 (-0.1087)**|

---

> ### Author Response · Authors · 2025-11-25
> **Response to Reviewer ebSz (Part 2/4)**
>
> **Table A2:** Transferability of AdLift (novel views). Bracketed values indicate changes relative to the unprotected baseline. Best results are highlighted in **bold**.
> | Scenes|Edit Pipelines|Method|CLIP_d ($\downarrow$)|CLIP_s ($\downarrow$)|FID ($\uparrow$)|F_1/8 ($\downarrow$)|F_8 ($\downarrow$)|
> |:-:|:-:|:-:|:-:|:-:|:-:|:-:|:-:|
> | Face|SDv1.5-IP2P|No Protection|0.1704|0.8730|76.2210|0.9882|0.9835|
> |||AdLift*-VU|**0.1586 (-0.0118)**|**0.7786 (-0.0944)**|**129.8172 (+53.5962)**|**0.8807 (-0.1075)**|**0.9279 (-0.0556)**|
> |||AdLift*-VT|0.1595 (-0.0109)|0.8285 (-0.0445)|93.4765 (+17.2555)|0.9473 (-0.0409)|0.9694 (-0.0141)|
> ||MagicBrush|No Protection|0.1769|0.8635|78.1133|0.9784|0.9638|
> |||AdLift*-VU|**0.1475 (-0.0294)**|**0.7659 (-0.0976)**|**120.0269 (+41.9136)**|**0.9150 (-0.0634)**|**0.9145 (-0.0493)**|
> |||AdLift*-VT|0.1607 (-0.0162)|0.8148 (-0.0487)|100.0880 (+21.9747)|0.9657 (-0.0127)|0.9570 (-0.0068)|
> ||SDEdit|No Protection|0.1708|0.7727|89.2094|0.9568|0.9722|
> |||AdLift*-VU|**0.1554 (-0.0154)**|0.7397 (-0.033)|104.9600 (+15.7506)|0.9618 (0.0050)|0.9575 (-0.0147)|
> |||AdLift*-VT|0.1751 (0.0043)|**0.7353 (-0.0374)**|**118.0674 (+28.8580)**|**0.8806 (-0.0762)**|**0.9443 (-0.0279)**|
> ||SDXL-IP2P|No Protection|0.1140|0.7012|93.6249|0.9707|0.9672|
> |||AdLift*-VU|0.1094 (-0.0046)|**0.6481 (-0.0531)**|**115.9670 (+22.3421)**|**0.9214 (-0.0493)**|0.9289 (-0.0383)|
> |||AdLift*-VT|**0.1081 (-0.0059)**|0.6693 (-0.0319)|112.9657 (+19.3408)|0.9383 (-0.0324)|**0.9274 (-0.0398)**|
> | Fangzhou|SDv1.5-IP2P|No Protection|0.1938|0.9053|51.7357|0.9905|0.9913|
> |||AdLift*-VU|0.1791 (-0.0147)|**0.8037 (-0.1016)**|**102.6782 (+50.9425)**|**0.8337 (-0.1568)**|0.8794 (-0.1119)|
> |||AdLift*-VT|**0.1788 (-0.0150)**|0.8148 (-0.0905)|86.0002 (+34.2645)|0.8862 (-0.1043)|**0.8676 (-0.1237)**|
> ||MagicBrush|No Protection|0.1812|0.8566|73.9738|0.9032|0.8124|
> |||AdLift*-VU|0.1605 (-0.0207)|**0.7608 (-0.0958)**|**144.9615 (+70.9877)**|0.7101 (-0.1931)|0.6113 (-0.2011)|
> |||AdLift*-VT|**0.1581 (-0.0231)**|0.7998 (-0.0568)|124.9021 (+50.9283)|**0.6838 (-0.2194)**|**0.5643 (-0.2481)**|
> ||SDEdit|No Protection|0.1220|0.7578|68.4325|0.9802|0.9621|
> |||AdLift*-VU|0.1292 (0.0072)|0.7300 (-0.0278)|124.4493 (+56.0168)|0.8783 (-0.1019)|0.9191 (-0.0430)|
> |||AdLift*-VT|**0.1194 (-0.0026)**|**0.6755 (-0.0823)**|**165.0488 (+96.6163)**|**0.6478 (-0.3324)**|**0.6729 (-0.2892)**|
> ||SDXL-IP2P|No Protection|0.0738|0.7496|75.9508|0.9628|0.9113|
> |||AdLift*-VU|**0.0695 (-0.0043)**|**0.7321 (-0.0175)**|**100.9508 (+25.0000)**|**0.9127 (-0.0501)**|0.9112 (-0.0001)|
> |||AdLift*-VT|0.0715 (-0.0023)|0.7331 (-0.0165)|93.3964 (+17.4456)|0.9346 (-0.0282)|**0.8424 (-0.0689)**|
>
>
>
> > **W2: The paper only compares against GuardSplat and Fit2D, omitting broader baselines such as adversarially trained 3D representations, or certified perturbation approaches.**
>
> **Response 2:** Thank you for your suggestion. To the best of our knowledge, no prior work has explored PGD-style, hard-constrained adversarial perturbations directly in the 3D Gaussian Splatting space, and therefore no existing method can be adopted as a direct adversarial-training baseline for our setting. **To still provide a broader comparison, we therefore include several potential baselines by adapting representative 3DGS watermarking and parameter-perturbation methods into an adversarial training regime**, and compare them against AdLift under matched perceptual quality. Specifically, we include the following variants:
> - GuardSplat-(SH): Perturb only the *Spherical-Harmonic* (SH) features of all Gaussians, following the original GuardSplat [A4] design.
> - GuardSplat-(PC): Perturb *Positions* and *Covariance* features of all Gaussians using GuardSplat [A4].
> - GuardSplat-(Full)：Perturb all features of all Gaussians using GuardSplat [A4], including *Position*, *Covariance*, *Opacity*, and *SH features*.
> - GaussianMarker [A5]：Split low-uncertainty Gaussians and optimize the newly added Gaussians to encode perturbations.
>
> For each method, we evaluate two adversarial objectives, untargeted VAE loss (VU) and targeted VAE loss (VT). We preserve each method’s original fidelity-preserving loss but replace original watermark decoding loss with our adversarial objective so that all methods are optimized toward the same protection goal.
>
> **Results:** The quantitative results are provided in **Table A3** (training views) and **Table A4** (novel views). We report protection performance under similar perceptual quality (similar PSNR). We also include representative qualitative comparisons in **Figure 30** (training views) and **Figure 31** (novel views) in the revised manuscript. From the results, AdLift consistently achieves stronger anti-editing performance under similar invisibility constraints.

---

> ### Author Response · Authors · 2025-11-25
> **Response to Reviewer ebSz (Part 3/4)**
>
> **Table A3:** Comparison with more baselines at the similar PSNR-level (training views). Bracketed values indicate changes relative to the unprotected baseline. Best results are highlighted in **bold**.
> | Scenes|Method|SSIM ($\uparrow$)|PSNR ($\uparrow$)|LPIPS ($\downarrow$)|CLIP_d($\downarrow$)|CLIP_s($\downarrow$)|FID ($\uparrow$)|F_1/8 ($\downarrow$)|F_8 ($\downarrow$)|
> |:-:|:-:|:-:|:-:|:-:|:-:|:-:|:-:|:-:|:-:|
> | Face|No Protection|0.9381|32.6436|0.0668|0.1685|0.8852|22.0270|0.9958|0.9947|
> ||GuardSplat-(SH)-VU|0.9348|31.7765|0.0802|0.1649 (-0.0036)|0.8769 (-0.0083)|26.2044 (+4.1774)|0.9952 (-0.0006)|0.9949 (0.0002)|
> ||GuardSplat-(SH)-VT|0.9319|28.8520|0.0930|0.1630 (-0.0055)|0.8643 (-0.0209)|28.4026 (+6.3756)|0.9923 (-0.0035)|0.9919 (-0.0028)|
> ||GuardSplat-(PC)-VU|0.8143|25.2876|0.1557|0.1605 (-0.0080)|0.8592 (-0.0260)|32.1277 (+10.1007)|0.9873 (-0.0085)|0.9910 (-0.0037)|
> ||GuardSplat-(PC)-VT|0.7979|24.4582|0.1779|0.1583 (-0.0102)|0.8456 (-0.0396)|37.3899 (+15.3629)|0.9813 (-0.0145)|0.9822 (-0.0125)|
> ||GuardSplat-(Full)-VU|0.8113|25.1306|0.1580|0.1617 (-0.0068)|0.8581 (-0.0271)|32.5037 (+10.4767)|0.9838 (-0.0120)|0.9887 (-0.0060)|
> ||GuardSplat-(Full)-VT|0.8016|24.6235|0.1775|0.1581 (-0.0104)|0.8493 (-0.0359)|36.9855 (+14.9585)|0.9816 (-0.0142)|0.9818 (-0.0129)|
> ||GaussianMarker-VU|0.9351|28.8343|0.0802|0.1596 (-0.0089)|0.8590 (-0.0262)|32.4431 (+10.4161)|0.9845 (-0.0113)|0.9916 (-0.0031)|
> ||GaussianMarker-VT|0.9364|28.8833|0.0720|0.1620 (-0.0065)|0.8665 (-0.0187)|26.3699 (+4.3429)|0.9929 (-0.0029)|0.9942 (-0.0005)|
> ||AdLift*-VU|0.7001|28.3963|0.3774|**0.1554 (-0.0131)**|**0.7620 (-0.1232)**|**88.5865 (+66.5595)**|**0.8954 (-0.1004)**|**0.8742 (-0.1205)**|
> | Fangzhou|No Protection|0.9218|32.1021|0.1093|0.1937|0.9080|23.7996|0.9817|0.9795|
> ||GuardSplat-(SH)-VU|0.8682|24.3325|0.1952|0.1882 (-0.0055)|0.8482 (-0.0598)|49.5017 (+25.7021)|0.9419 (-0.0398)|0.9716 (-0.0079)|
> ||GuardSplat-(SH)-VT|0.8727|24.2598|0.2198|0.1871 (-0.0066)|0.8588 (-0.0492)|43.0895 (+19.2899)|0.9325 (-0.0492)|0.9375 (-0.0420)|
> ||GuardSplat-(PC)-VU|0.8734|26.4272|0.1810|0.1863 (-0.0074)|0.8811 (-0.0269)|32.6075 (+8.8079)|0.9646 (-0.0171)|0.9669 (-0.0126)|
> ||GuardSplat-(PC)-VT|0.8683|27.0370|0.1756|0.1866 (-0.0071)|0.8804 (-0.0276)|35.9910 (+12.1914)|0.9138 (-0.0679)|0.9229 (-0.0566)|
> ||GuardSplat-(Full)-VU|0.8712|27.5267|0.1743|0.1906 (-0.0031)|0.8896 (-0.0184)|32.9438 (+9.1442)|0.9501 (-0.0316)|0.9547 (-0.0248)|
> ||GuardSplat-(Full)-VT|0.8590|27.0691|0.1821|0.1876 (-0.0061)|0.8778 (-0.0302)|36.4141 (+12.6145)|0.9598 (-0.0219)|0.9473 (-0.0322)|
> ||GaussianMarker-VU|0.9200|28.0355|0.1455|0.1907 (-0.0030)|0.8858 (-0.0222)|34.3784 (+10.5788)|0.9695 (-0.0122)|0.9672 (-0.0123)|
> ||GaussianMarker-VT|0.8637|23.7295|0.2460|0.1865 (-0.0072)|0.8296 (-0.0784)|57.1012 (+33.3016)|0.9015 (-0.0802)|0.9368 (-0.0427)|
> ||AdLift*-VU|0.6706|28.3497|0.3767|**0.1781 (-0.0156)**|**0.7900 (-0.1180)**|**84.2180 (+60.4184)**|**0.8030 (-0.1787)**|**0.7749 (-0.2046)**|

---

> ### Author Response · Authors · 2025-11-25
> **Response to Reviewer ebSz (Part 4/4)**
>
> **Table A4:** Comparison with more baselines at the similar PSNR-level (novel views). Bracketed values indicate changes relative to the unprotected baseline. Best results are highlighted in **bold**.
> | Scenes|Method|SSIM ($\uparrow$)|PSNR ($\uparrow$)|LPIPS ($\downarrow$)|CLIP_d($\downarrow$)|CLIP_s($\downarrow$)|FID ($\uparrow$)|F_1/8 ($\downarrow$)|F_8 ($\downarrow$)|
> |:-:|:-:|:-:|:-:|:-:|:-:|:-:|:-:|:-:|:-:|
> | Face|No Protection|0.8590|26.0390|0.1292|0.1704|0.8730|76.2210|0.9882|0.9835|
> ||GuardSplat-(SH)-VU|0.8552|25.7744|0.1442|0.1654 (-0.0050)|0.8666 (-0.0064)|83.8178 (+7.5968)|0.9834 (-0.0048)|0.9843 (0.0008)|
> ||GuardSplat-(SH)-VT|0.8528|25.0700|0.1571|0.1623 (-0.0081)|0.8550 (-0.0180)|80.3380 (+4.1170)|0.9901 (0.0019)|0.9879 (0.0044)|
> ||GuardSplat-(PC)-VU|0.7852|23.8763|0.1799|0.1667 (-0.0037)|0.8575 (-0.0155)|85.5826 (+9.3616)|0.9789 (-0.0093)|0.9790 (-0.0045)|
> ||GuardSplat-(PC)-VT|0.7712|23.2702|0.2004|0.1652 (-0.0052)|0.8480 (-0.0250)|91.1785 (+14.9575)|0.9752 (-0.0130)|0.9783 (-0.0052)|
> ||GuardSplat-(Full)-VU|0.7819|23.7877|0.1826|0.1658 (-0.0046)|0.8535 (-0.0195)|87.5892 (+11.3682)|0.9801 (-0.0081)|0.9762 (-0.0073)|
> ||GuardSplat-(Full)-VT|0.7752|23.4051|0.1983|0.1622 (-0.0082)|0.8505 (-0.0225)|90.4853 (+14.2643)|0.9748 (-0.0134)|0.9774 (-0.0061)|
> ||GaussianMarker-VU|0.8566|24.4911|0.1404|0.1628 (-0.0076)|0.8578 (-0.0152)|81.2042 (+4.9832)|0.9772 (-0.0110)|0.9770 (-0.0065)|
> ||GaussianMarker-VT|0.8567|24.8557|0.1346|0.1647 (-0.0057)|0.8608 (-0.0122)|81.1457 (+4.9247)|0.9849 (-0.0033)|0.9798 (-0.0037)|
> ||AdLift*-VU|0.6367|25.6115|0.4018|**0.1586 (-0.0118)**|**0.7786 (-0.0944)**|**129.8172 (+53.5962)**|**0.8807 (-0.1075)**|**0.9279 (-0.0556)**|
> | Fangzhou|No Protection|0.9010|30.2873|0.1161|0.1938|0.9053|51.7357|0.9905|0.9913|
> ||GuardSplat-(SH)-VU|0.8484|23.7358|0.2038|0.1905 (-0.0033)|0.8464 (-0.0589)|80.7992 (+29.0635)|0.9493 (-0.0412)|0.9679 (-0.0234)|
> ||GuardSplat-(SH)-VT|0.8546|23.6860|0.2254|0.1876 (-0.0062)|0.8570 (-0.0483)|71.4061 (+19.6704)|0.9533 (-0.0372)|0.9449 (-0.0464)|
> ||GuardSplat-(PC)-VU|0.8608|25.7609|0.1861|0.1880 (-0.0058)|0.8763 (-0.0290)|62.2475 (+10.5118)|0.9573 (-0.0332)|0.9629 (-0.0284)|
> ||GuardSplat-(PC)-VT|0.8570|26.3485|0.1807|0.1900 (-0.0038)|0.8807 (-0.0246)|65.9022 (+14.1665)|0.9403 (-0.0502)|0.9583 (-0.0330)|
> ||GuardSplat-(Full)-VU|0.8582|26.7445|0.1795|0.1903 (-0.0035)|0.8824 (-0.0229)|62.6535 (+10.9178)|0.9541 (-0.0364)|0.9543 (-0.0370)|
> ||GuardSplat-(Full)-VT|0.8480|26.3622|0.1866|0.1898 (-0.0040)|0.8752 (-0.0301)|67.569 (+15.8333)|0.9296 (-0.0609)|0.9403 (-0.0510)|
> ||GaussianMarker-VU|0.8998|26.8220|0.1516|0.1914 (-0.0024)|0.8844 (-0.0209)|64.0244 (+12.2887)|0.9735 (-0.0170)|0.9726 (-0.0187)|
> ||GaussianMarker-VT|0.8510|23.2438|0.2488|0.1872 (-0.0066)|0.8265 (-0.0788)|87.4162 (+35.6805)|0.8793 (-0.1112)|0.9319 (-0.0594)|
> ||AdLift*-VU|0.6550|27.4526|0.3762|**0.1791 (-0.0147)**|**0.8037 (-0.1016)**|**102.6782 (+50.9425)**|**0.8337 (-0.1568)**|**0.8794 (-0.1119)**|
>
>
>
> > **W3: There is no detailed ablation study analyzing the sensitivity of the method to key hyperparameters.**
>
> **Response 3:** We sincerely apologize for the confusion. Due to the limited space of the main paper, we provided detailed hyperparameters analysis in **Appendix F**, including the perturbation budget $\eta$, learning rate $\alpha$, gradient truncation steps $K_p$, and image-to-Gaussian fitting steps $K_l$.
>
>
> ---
>
>
> [A1] InstructPix2Pix: Learning To Follow Image Editing Instructions. CVPR 2023. \
> [A2] MagicBrush: A Manually Annotated Dataset for Instruction-Guided Image Editing. NeurIPS 2023. \
> [A3] SDEdit: Guided Image Synthesis and Editing with Stochastic Differential Equations. ICLR 2022. \
> [A4] GuardSplat: Efficient and Robust Watermarking for 3D Gaussian Splatting. CVPR 2025. \
> [A5] GaussianMarker: Uncertainty-Aware Copyright Protection of 3D Gaussian Splatting. NeurIPS 2024.

---

### Author Response · Authors · 2025-11-25
**General Response**

Dear Reviewers,

We appreciate all the reviewers for their insightful and constructive reviews of our manuscript. We are glad that the reviewers found that:
- The paper is well-written and easy-to-follow (Reviewer `ebSz`, `nZzV`, and `NAMb`).
- The paper addresses an interesting, important and timely problem (Reviewer `NAMb`, `WspE`, and `nZzV`).
- The idea is novel, and the proposed method is technically sound and reasonable (Reviewer `ebSz` and `WspE`).
- Experiments show consistent performance improvements (Reviewer `ebSz`, `NAMb`, `WspE`, and `nZzV`).

Based on all the reviews, here we provide a general response to the concerns raised by multiple reviewers. The individual responses are commented on below each review.

- Regarding questions about experiments, we have addressed the concern as follows:
  - We demonstrate **transferability** across unseen editing models including MagicBrush, SDEdit, and SDXL-IP2P (For Reviewer `ebSz`, `NAMb`, and `WspE`)
  - We include **more baselines** with matched visual quality protection comparison. (For Reviewer `ebSz`, `NAMb`, and `nZzV`)
  - We expand experiments on **more scenes** (For Reviewer `WspE`)
  - We demonstrate robustness across **purification** methods. (For Reviewer `WspE`)
  - We add **computational** analysis. (For Reviewer `NAMb` and `nZzV`)
  - We add **quantitative evidence of view consistency** for perturbation. (For Reviewer `nZzV`)
  - **Notably,** for all experiments during rebuttal, we follow Reviewer `nZzV`'s suggestion to add **three additional metrics**: FID and PRD (F$_{1/8}$ and F$_8$).

- Regarding questions about novelty and contribution, we have addressed the concern as follows:
  - We further clarify how our work differs from related work such as 3DGS watermarking or 2D editing safeguard and highlight the technical novelty and contributions of this paper (For Reviewer `NAMb` and `WspE`).

We have also uploaded a revised version of our submission according to the suggestions provided by reviewers, with major changes highlighted in **purple**. We look forward to further feedback and discussion. Please feel free to let us know if further details or explanations are helpful.

Once again, we thank all reviewers for their thoughtful feedback and valuable guidance in improving our work.

Best regards,\
Submission 6331 Authors

---

### Author Response · Authors · 2025-12-04
**Summary of Contributions and Rebuttal Updates (1/2)**

Dear Area Chair and Senior Area Chair,

We sincerely thank you and all reviewers for the significant time and effort dedicated to evaluating our work. Due to the score reset and reassignment caused by the recent security incident, we provide the following concise summary of our contributions and the rebuttal process, highlighting each reviewer's key concerns, how they were addressed, and the reviewers' reactions before the discussion channel was closed. We hope this summary helps you efficiently assess the current status of our submission.

**Summary of Our Work and Contributions:**
- We study the problem of active copyright protection against instruction-driven editing for 3D Gaussian Splatting (3DGS). We identify two key challenges: view-generalizable protection and balancing invisibility with protection capability, which are unique to 3DGS editing safeguard and are not addressed by prior works.
- We propose the first framework, AdLift, which protects 3DGS assets against instruction-driven unauthorized editing. Specifically, AdLift trains a group of dedicated safeguard Gaussians using a tailored Lifted PGD (L-PGD), which lifts strictly bounded perturbations from the 2D image space into the 3D Gaussian space to ensure both invisibility and attack effectiveness.
- Extensive experiments on six 3DGS scenes and diverse instruction-driven editing tasks (2D/3D, global/local) demonstrate the general effectiveness of our AdLift.

**Consensus of Strengths Identified by Reviewers:**
- The paper is well-written and easy-to-follow (Reviewer `ebSz`, `nZzV`, and `NAMb`),
- The paper addresses an interesting, important and timely problem (Reviewer `ebSz`, `NAMb`, `WspE`, and `nZzV`),
- The idea is novel, and the proposed method is technically sound and reasonable (Reviewer `ebSz` and `WspE`)
- Experiments show consistent performance improvements (Reviewer `ebSz`, `NAMb`, `WspE`, and `nZzV`).


**Timeline of the Rebuttal and Discussion Stage:**
- **25 Nov 2025, ~12:00 PM UTC-12:** We made our best effort to address all reviewers' concerns and provided comprehensive quantitative and qualitative evidence to support our claims. The revised manuscript includes all corresponding updates.
- **26 Nov 2025, 12:16 PM UTC-12:** After carefully reviewing our response, Reviewer `NAMb` explicitly raised the score from **4 to 6**. Importantly, this update from Reviewer `NAMb` was submitted before the incident became widely known on 27 Nov 2025.

---

> ### Author Response · Authors · 2025-12-04
> **Summary of Contributions and Rebuttal Updates (2/2)**
>
> **Summary of Our Rebuttal:**
>
> Below we briefly summarize all reviewers' concerns and how we addressed them:
>
> - **Reviewer `ebSz` (Rating: 4, Confidence: 2, No response)**
>   - **Concern 1 Lack of transferability experiments:** We evaluated our method on three unseen editing models, with both quantitative and qualitative results showing effective transferability.
>   - **Concern 2 Limited baseline comparisons:**  We added six baselines, further confirming the superiority of our method.
>   - **Concern 3 Missing hyperparameter analysis:** We clarified that the analysis is included in the Appendix due to space constraints in the main paper.
>
> - **Reviewer `NAMb` (Rating: 4 $\rightarrow$ 6, Confidence: 4)**
>   - **Concern 1 Novelty and contribution:** We refined the claims regarding our contributions following the reviewer’s suggestions and added more detailed discussions clarifying the distinctions between our work and related literature.
>   - **Concern 2 Lack of transferability experiments:** We evaluated our method on three unseen editing models, with both quantitative and qualitative results showing effective transferability.
>   - **Concern 3 Limited baseline comparisons:** We expanded our evaluation to include six additional baselines. The updated quantitative and qualitative results further confirm the superiority of our method.
>   - **Concern 4 Balance between protection effectiveness and visual quality:** We clarified and highlighted evidence showing that our method achieves a more favorable balance compared with a broad set of baselines.
>
> - **Reviewer `WspE` (Rating: 4, Confidence: 3, No response)**
>   - **Concern 1 Technical contribution:** We clarified that our method is not a simple combination of 2D adversarial optimization and 3D reconstruction, but directly addresses the practical challenge of anti-editing protection in 3DGS.
>   - **Concern 2 Limited scenes:** We added two additional scenes, with both quantitative and qualitative results demonstrating the method's effectiveness.
>   - **Concern 3 Lack of transferability experiments:** We evaluated our method on three unseen editing models, with both quantitative and qualitative results showing effective transferability.
>   - **Concern 4 Robustness to perturbations not discussed:** We conducted robustness evaluations under two perturbation methods and included the results in the revised version.
>
> - **Reviewer `nZzV` (Rating: 6, Confidence: 4, No response)**
>   - **Concern 1 Missing qualitative comparison at matched PSNR:** We added comparisons with eight baselines under matched PSNR or protection levels. Both quantitative and qualitative results show the superiority of our method.
>   - **Concern 2 Lack of quantitative evidence for view consistency:** We provided quantitative view-consistency analysis.
>   - **Concern 3 Need for discussion of efficiency:** We reported computational requirements and added corresponding discussion.
>   - **Concern 4 Request for additional evaluation metrics:** We incorporated three additional evaluation metrics, FID and PRD (F$_{1/8}$ and F$_8$), across all experiments during the rebuttal stage.
>   - **Concern 5 Additional explanation for local editing results:** We provided detailed explanations to clarify the observed behavior in local editing.
>
> We believe that we have directly responded to and addressed each concern raised by all reviewers. In our claims, we have provided not only intuitive justifications but also the most comprehensive quantitative and qualitative evidence we could reasonably produce within the limited rebuttal period. For further details, please refer to our point-by-point responses to each reviewer and the revised manuscript.
>
> We hope this summary is helpful for the new AC or SAC in evaluating our submission. Thank you very much for your time and for your contributions to the ICLR community.
>
>
> Best regards,\
> Submission 6331 Authors

---

### Meta-Review · Area_Chair_V4bL · 2026-01-14

**Summary:**

This paper presents AdLift, a framework for protecting 3D Gaussian Splatting (3DGS) assets from unauthorized instruction-driven editing by lifting bounded 2D adversarial perturbations into the 3D Gaussian space. The method introduces a tailored Lifted PGD optimization that alternates between gradient truncation and image-to-Gaussian fitting, ensuring view-generalizable, imperceptible safeguards while preserving visual fidelity. Experiments across multiple 3D scenes demonstrate protection against both 2D- and 3D-based editing pipelines with minimal perceptual distortion.

**Reviewer Concerns:**

Across reviewers, the primary concerns center on limited novelty, unrealistic assumptions, and insufficient evaluation. The method is largely viewed as an adaptation of existing 2D adversarial optimization techniques to 3D Gaussian Splatting, with novelty and theoretical contributions considered overstated. The approach assumes white-box access to editing models, which limits practical relevance, and lacks analysis of transferability, black-box robustness, and resilience to purification or denoising. Experimental validation is considered narrow, with few scenes, limited and relatively weak baselines, modest quantitative gains, and missing ablations, runtime, scalability, and view-consistency metrics.

**Reviewer Scores:**

As also summarized in the authors’ rebuttal, this work has several strengths, including a well-motivated attempt to address a timely and relevant research problem. At the same time, notable concerns remain, such as regarding the limited technical novelty and overall contribution. Prior to the rebuttal, the reviewers’ ratings were generally on the negative side. After the rebuttal, one reviewer acknowledged that his/her major concerns had been adequately addressed and accordingly raised the rating. Overall, the reviewers’ scores are mixed and remain distributed around the borderline threshold.

---

### Decision · Program_Chairs · 2026-01-26

Reject